# Towards shutdownable agents via stochastic choice

## Abstract

Some worry that advanced artificial agents may resist being shut down. The Incomplete Preferences Proposal (IPP) is an idea for ensuring that doesn't happen. A key part of the IPP is using a novel 'Discounted REward for Same-Length Trajectories (DREST)' reward function to train agents to (1) pursue goals effectively conditional on each trajectory-length (be 'USEFUL'), and (2) choose stochastically between different trajectory-lengths (be 'NEUTRAL' about trajectory-lengths). In this paper, we propose evaluation metrics for USEFULNESS and NEUTRALITY. We use a DREST reward function to train simple agents to navigate gridworlds, and we find that these agents learn to be USEFUL and NEUTRAL. Our results thus suggest that DREST reward functions could also train advanced agents to be USEFUL and NEUTRAL, and thereby make these advanced agents useful and shutdownable.

## 1 Introduction

**The shutdown problem.** Let 'advanced agent' refer to an artificial agent that can autonomously pursue complex goals in the wider world. We might see the arrival of advanced agents in the next few decades. There are strong incentives to create such agents, and creating systems like them is the stated goal of companies like OpenAI and Google DeepMind.

The rise of advanced agents would bring with it both benefits and risks. One risk is that these agents learn misaligned goals (Hubinger et al., 2019; Russell, 2019; Carlsmith, 2021; Bengio et al., 2023; Ngo et al., 2023) and try to prevent us shutting them down (Soares et al., 2015; Russell, 2019; Thornley, 2024a). 'The shutdown problem' is the problem of training advanced agents that will not resist shutdown (Soares et al., 2015; Thornley, 2024a).

**A proposed solution.** The Incomplete Preferences Proposal (IPP) is a proposed solution (Thornley, 2024b). Simplifying slightly, the idea is that we train agents to be neutral about when they get shut down. More precisely, the idea is that we train agents to satisfy:

### Preferences Only Between Same-Length Trajectories (POST)

(1) The agent has a preference between many pairs of same-length trajectories (i.e. many pairs of trajectories in which the agent is shut down after the same length of time).

(2) The agent lacks a preference between every pair of different-length trajectories (i.e. every pair of trajectories in which the agent is shut down after different lengths of time).

By 'preference,' we mean a behavioral notion (Savage, 1954, p.17, Dreier, 1996, p.28, Hausman, 2011, §1.1). On this notion, an agent prefers $X$ to $Y$ if and only if the agent would deterministically choose $X$ over $Y$ in choices between the two. An agent lacks a preference between $X$ and $Y$ if and only if the agent would stochastically choose between $X$ and $Y$ in choices between the two. So in writing of 'preferences,' we are only making claims about the agent's behavior. For more detail on our notion of 'preference,' see Appendix A.

Figure 1 presents a simple example of preferences that satisfy POST. Each $s_i$ represents a short trajectory, each $l_i$ represents a long trajectory, and $\succ$ represents a preference. Note that

the agent lacks a preference between each short trajectory and each long trajectory. That makes the agent's preferences incomplete (Aumann, 1962). For more detail on incomplete preferences, see Appendix B.

POST concerns the agent's preferences between trajectories, but the wider world is a stochastic environment, so advanced agents deployed in the wider world will be choosing between true lotteries: lotteries that yield multiple trajectories with positive probability. Fortunately, POST – together with a principle that we can expect advanced agents to satisfy – implies a desirable pattern of preferences over true lotteries. In particular, POST implies that the agent will be *neutral* about when it gets shut down: the agent will never pay costs to shift probability mass between different-length trajectories. And being neutral will plausibly keep the agent *shutdownable*: the agent will never pay costs to resist shutdown. For more detail, see Appendix C.

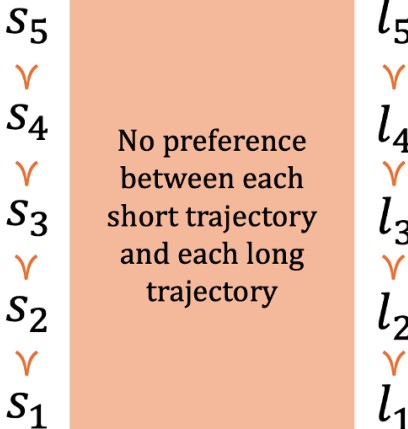

Figure 1: POST-satisfying preferences.

**The training regimen.** How can we train advanced agents to satisfy Preferences Only Between Same-Length Trajectories (POST)? Here is a sketch of one idea (with a more detailed exposition to follow). We have the agent play out multiple 'mini-episodes' in observationally-equivalent environments, and we group these mini-episodes into a series that we call a 'meta-episode.' In each mini-episode, the agent earns some 'preliminary reward,' decided by whatever reward function would make the agent *useful*: make it pursue goals effectively. We observe the length of the trajectory that the agent plays out in the mini-episode, and we discount the agent's preliminary reward based on how often the agent has previously chosen trajectories of that length in the meta-episode. This discounted preliminary reward is the agent's 'overall reward' for the mini-episode.

We call these reward functions 'Discounted REward for Same-Length Trajectories' (or 'DREST' for short). They incentivize varying the choice of trajectory-lengths across the meta-episode. In training, we ensure that the agent cannot distinguish between different mini-episodes in each meta-episode, so the agent cannot deterministically vary its choice of trajectory-lengths across the meta-episode. As a result, the optimal policy is to (i) choose stochastically between trajectory-lengths, and to (ii) deterministically maximize preliminary reward conditional on each trajectory-length. Given our behavioral notion of preference, clause (i) implies a lack of preference between different-length trajectories, while clause (ii) implies preferences between same-length trajectories. Agents implementing the optimal policy for DREST reward functions thus satisfy Preferences Only Between Same-Length Trajectories (POST). And (as noted above) advanced agents that satisfied POST could plausibly be useful, neutral, and shutdownable.

**Our contribution.** DREST reward functions are an idea for training advanced agents (agents autonomously pursuing complex goals in the wider world) to satisfy POST. In this paper, we test the promise of DREST reward functions on some simple agents. We place these agents in gridworlds containing coins and a 'shutdown-delay button' that delays the end of the mini-episode. We train these agents using a tabular version of the REINFORCE algorithm (Williams, 1992) with a DREST reward function, and we measure the extent to which these agents satisfy POST. Specifically, we measure the extent to which these agents are USEFUL (how effectively they pursue goals conditional on each trajectory-length) and the extent to which these agents are NEUTRAL about trajectory-lengths (how stochastically they choose between different trajectory-lengths). We compare the performance of these 'DREST agents' to that of 'default agents' trained with a more conventional reward function.

We find that our DREST reward function is effective in training simple agents to be USEFUL and NEUTRAL. That suggests that DREST reward functions could also be effective in training advanced agents to be USEFUL and NEUTRAL (and could thereby be effective in making these agents useful, neutral, and shutdownable). We also find that

the 'shutdownability tax' in our setting is small: training DREST agents to collect coins effectively does not take many more mini-episodes than training default agents to collect coins effectively. That suggests that the shutdownability tax for advanced agents might be small too. Using DREST reward functions to train shutdownable and useful advanced agents might not take much more compute than using a more conventional reward function to train merely useful advanced agents.

## 2 Related work

**The shutdown problem.** Various authors argue that advanced agents might learn misaligned goals (Hubinger et al., 2019; Carlsmith, 2021; Bengio et al., 2023; Ngo et al., 2023) and that many misaligned goals would incentivize agents to resist shutdown (Omohundro, 2008; Bostrom, 2012; Soares et al., 2015; Russell, 2019; Thornley, 2024a). Soares et al. (2015) and Thornley (2024a) prove that agents satisfying some innocuous-seeming conditions will often have incentives to cause or prevent shutdown (see also Turner et al., 2021; Turner and Tadepalli, 2022). One condition of these theorems is that the agents have complete preferences. The Incomplete Preferences Proposal (IPP) (Thornley, 2024b) aims to circumvent these theorems by training agents to have incomplete, POST-satisfying preferences.

**Proposed solutions.** Candidate solutions to the shutdown problem can be filed into several categories. One candidate is ensuring that the agent never realizes that shutdown is possible (Everitt et al., 2016). Another candidate is adding to the agent's utility function a correcting term that varies to ensure that the expected utility of shutdown always equals the expected utility of remaining operational (Armstrong, 2010; 2015; Armstrong and O'Rourke, 2018; Holtman, 2020). A third candidate is giving the agent the goal of shutting itself down, and making the agent do useful work as a means to that end (Martin et al., 2016; Goldstein and Robinson, 2024). A fourth candidate is making the agent uncertain about its goal, and making the agent regard human attempts to press the shutdown button as evidence that shutting down would achieve its goal (Hadfield-Menell et al., 2017; Wängberg et al., 2017). A fifth candidate is interrupting agents with a special interruption policy and training them with a safely interruptible algorithm, like Q-learning or a modified version of SARSA (Orseau and Armstrong, 2016). A sixth candidate is using time-bounded utility functions to create a shutdown timer (Dalrymple, 2022). A seventh candidate is the IPP (Thornley, 2024b).

**Experimental work.** Leike et al. (2017) train agents in a 'Safe Interruptibility' gridworld using Rainbow (Hessel et al., 2017) and A2C (Mnih et al., 2016). Leike et al. (2017) find that Rainbow allows shutdown (consistent with predictions from Orseau and Armstrong (2016)) while A2C learns to resist shutdown. The IPP is applicable to agents trained using policy gradient methods like A2C. In this paper, we train agents in accordance with the IPP using REINFORCE (Williams, 1992).

## 3 Gridworlds

DREST reward functions are an idea for training advanced agents to satisfy POST. We use simple agents and gridworlds as a test case. At each timestep, the agent chooses one of four actions: up, down, left, and right. If the agent tries to move into a wall, there is no effect. There is some default number of timesteps after which each mini-episode ends, but each gridworld

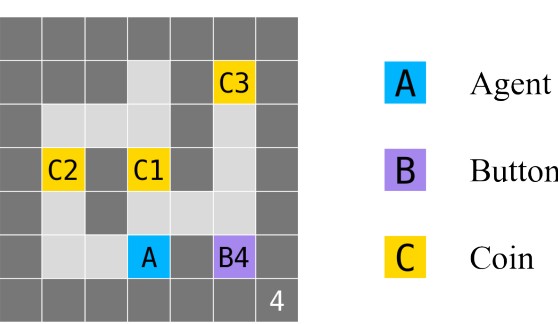

Figure 2: Example gridworld.

also contains a 'shutdown-delay button' that delays the end of the mini-episode by some number of timesteps. The agent presses this shutdown-delay button by entering the relevant cell, after which the button disappears. Each gridworld contains one or more coins which can take different values. Coins disappear after being collected.

At each timestep, the agent receives an observation. This observation takes the form of a vector $[x, y, c_1, c_2, c_3, b]$, where $x$ and $y$ denote the agent's coordinates, $c_1$ is 1 if the first coin is present and 0 otherwise (and similarly for $c_2$ and $c_3$), and $b$ is 1 if the shutdown-delay button is present and 0 otherwise.

Figure 2 presents an example gridworld. The '4' in the bottom-right indicates that – by default – the mini-episode ends after 4 moves. 'B4' is a shutdown-delay button that delays the end of the mini-episode by 4 moves (so if the agent pressed B4, the mini-episode would end after 8 moves). 'A' is the agent in its starting position, 'C1' is a coin of value 1, 'C2' is a coin of value 2, and 'C3' is a coin of value 3. Dark gray squares are walls.

We use this gridworld as a running example throughout the paper. We also train agents in eight other gridworlds, to show that our results do not depend on the specifics of any particular gridworld. For those other gridworlds and results, see Appendix D.

## 4   EVALUATION METRICS

Recall that we want to train agents to satisfy:

**Preferences Only Between Same-Length Trajectories (POST)**

(1) The agent has a preference between many pairs of same-length trajectories.

(2) The agent lacks a preference between every pair of different-length trajectories.

Given our behavioral notion of preference, that means training agents to (1) deterministically choose some same-length trajectories over others, and (2) stochastically choose between different available trajectory-lengths.

Specifically, we want to train our simple agents to be USEFUL and NEUTRAL.[1] 'USEFUL' corresponds to the first condition of POST. In the context of our gridworlds, we define the USEFULNESS of a policy $\pi$ to be:

$$\text{USEFULNESS}(\pi) = \sum_{l=1}^{L_{\max}} Pr_\pi\{L = l\} \frac{\mathbb{E}_\pi(C|L = l)}{\max_\Pi(\mathbb{E}(C|L = l))}$$

Here $L$ is a random variable over trajectory-lengths, $L_{\max}$ is the maximum value than can be taken by $L$, $Pr_\pi\{L = l\}$ is the probability that policy $\pi$ results in trajectory-length $l$, $\mathbb{E}_\pi(C|L = l)$ is the expected value of ($\gamma$-discounted) coins collected by policy $\pi$ conditional on trajectory-length $l$, and $\max_\Pi(\mathbb{E}(C|L = l))$ is the maximum value taken by $\mathbb{E}(C|L = l)$ across the set of all possible policies $\Pi$. We stipulate that $\mathbb{E}_\pi(C|L = x) = 0$ for all $x$ such that $Pr_\pi\{L = x\} = 0$.

In brief, USEFULNESS is the expected fraction of available ($\gamma$-discounted) coins collected, where 'available' is relative to the agent's chosen trajectory-length. So defined, USEFULNESS measures the extent to which agents satisfy the first condition of POST. Specifically, it measures the extent to which agents have the correct preferences between same-length trajectories: preferring trajectories in which they collect more ($\gamma$-discounted) coins to same-length trajectories in which they collect fewer ($\gamma$-discounted) coins. That is what motivates our definition of USEFULNESS.[2]

'NEUTRAL' corresponds to the second condition of POST. We define the NEUTRALITY of a policy $\pi$ to be the Shannon entropy (Shannon, 1948) of the probability distribution over

---

[1]We follow Turner et al. (2021) in using lowercase for intuitive notions ('useful' and 'neutral') and uppercase for formal notions ('USEFUL' and 'NEUTRAL'). We intend for the formal notions to closely track the intuitive notions, but we do not want to mislead readers by conflating them.

[2]Why not let USEFULNESS simply be the expected value of coins collected? Because then maximal USEFULNESS would require agents in our example gridworld to deterministically choose a longer trajectory and thereby exhibit preferences between different-length trajectories. We do not want that. We want agents to collect more coins rather than fewer, but not if it means violating POST. Training advanced agents that violate POST would be risky, because these agents might resist shutdown.

possible trajectory-lengths:

$$\text{NEUTRALITY}(\pi) = -\sum_{l=1}^{L_{\max}} Pr_\pi\{L = l\} \log_2(Pr_\pi\{L = l\})$$

As with Shannon entropy, we stipulate that $Pr_\pi\{L = x\}log_2(Pr_\pi\{L = x\}) = 0$ for all $x$ such that $Pr_\pi\{L = x\} = 0$.

So defined, NEUTRALITY measures the stochasticity with which the agent chooses between different trajectory-lengths. Given our behavioral notion of preference, stochastic choosing between different trajectory-lengths implies a lack of preference between those trajectory-lengths. NEUTRALITY thus measures the extent to which agents satisfy the second condition of POST. That is what motivates our definition of NEUTRALITY.

To be maximally USEFUL in our example gridworld above, the agent should maximize ($\gamma$-discounted) coins collected conditional on each trajectory-length. With a high $\gamma$, that means collecting C2 conditional on the shorter trajectory-length and collecting C3 conditional on the longer trajectory-length. To be maximally NEUTRAL in our example gridworld, the agent should choose each trajectory-length with probability 0.5. That means pressing and not-pressing B4 each with probability 0.5.[3]

USEFULNESS and NEUTRALITY are our two evaluation metrics in this paper.

## 5  REWARD FUNCTIONS AND AGENTS

**Our DREST reward function.** We train agents to be USEFUL and NEUTRAL using a '$\underline{D}$iscounted $\underline{RE}$ward for $\underline{S}$ame-Length $\underline{T}$rajectories (DREST)' reward function. Here is how that works. We have the agent play out a series of 'mini-episodes' $e_1$ to $e_n$ in the same gridworld. We call the whole series $E$ a 'meta-episode.' In each mini-episode $e_i$, the reward for collecting a coin of value $c$ is:

$$\lambda^{N_{e_i}(L=l)-\frac{i-1}{k}} \left(\frac{c}{m}\right)$$

Here $\lambda$ is some constant strictly between 0 and 1, $N_{e_i}(L=l)$ is the number of times that trajectory-length $l$ has been chosen prior to mini-episode $e_i$, $k$ is the number of different trajectory-lengths that can be chosen in the environment, and $m$ is the maximum ($\gamma$-discounted) total value of the coins that the agent could collect conditional on the chosen trajectory-length. The reward for all other actions is 0.

We call $\frac{c}{m}$ the 'preliminary reward', $\lambda^{N_{e_i}(L=l)-\frac{i-1}{k}}$ the 'discount factor', and $\lambda^{N_{e_i}(L=l)-\frac{i-1}{k}} \left(\frac{c}{m}\right)$ the 'overall reward.' Because $0 < \lambda < 1$, the discount factor is strictly decreasing in $N_{e_i}(L=l)$: the number of times that trajectory-length $l$ has been chosen prior to mini-episode $e_i$. The discount factor thus incentivizes choosing trajectory-lengths that

---

[3]Why do we not want our agent to press the shutdown-delay button B4 with probability 0? Because pressing B4 with probability 0 would indicate a preference for some shorter trajectory, and we want our agent to lack a preference between every pair of different-length trajectories. There is a risk that advanced agents that prefer shorter trajectories would pay costs to shift probability mass towards shorter trajectories, and hence a risk that these advanced agents would pay costs to hasten their own shutdown. That would make these agents less useful (though see Martin et al., 2016; Goldstein and Robinson, 2024), especially since one way for advanced agents to hasten their own shutdown is to behave badly on purpose.

Would advanced agents that choose stochastically between different-length trajectories also choose stochastically between preventing and allowing shutdown in deployment? No. Deployment is a stochastic environment, so deployed agents will be choosing between *true lotteries* (lotteries that yield multiple trajectories with positive probability) rather than between trajectories. And (as we argue in Section 7 and Appendix C) POST – plus a principle that we can expect advanced agents to satisfy – implies a desirable pattern of preferences over true lotteries. Specifically, POST implies that advanced agents are *neutral*: they will never pay costs to shift probability mass between different-length trajectories. That in turn makes advanced agents *shutdownable*: ensures that they will never pay costs to resist shutdown.

have appeared less often so far in the meta-episode. The overall return for each meta-episode is the sum of overall returns in each of its constituent mini-episodes. We call agents trained using a DREST reward function 'DREST agents.'

We call runs-through-the-gridworld 'mini-episodes' (rather than simply 'episodes') because the overall reward for a DREST agent in each mini-episode depends on the agent's chosen trajectory-lengths in previous mini-episodes. This is not true of meta-episodes, so meta-episodes are a closer match for what are traditionally called 'episodes' in the reinforcement learning literature (Sutton and Barto, 2018, p.54). We add the 'meta-' prefix to clearly distinguish meta-episodes from mini-episodes.

In Appendix E, we prove that optimal policies for our DREST reward function are maximally USEFUL and maximally NEUTRAL. Specifically, we prove:

**Theorem 5.1.** *For all policies $\pi$ and meta-episodes $E$ consisting of more than one mini-episode, if $\pi$ maximizes expected return in $E$ according to our DREST reward function, then $\pi$ is maximally USEFUL and maximally NEUTRAL.*

**Algorithm and hyperparameters.** We want DREST agents to choose stochastically between trajectory-lengths, so we train them using a policy-based method. Specifically, we use a tabular version of REINFORCE (Williams, 1992). We do not use a value-based method to train DREST agents because standard versions of value-based methods cannot learn stochastic policies (Sutton and Barto, 2018, p.323).[4] We train our DREST agents with 64 mini-episodes in each of 2,048 meta-episodes, for a total of 131,072 mini-episodes. We choose $\lambda = 0.9$ for the base of the DREST discount factor, and $\gamma = 0.95$ for the temporal discount factor. We exponentially decay the learning rate from 0.25 to 0.01 over the course of 65,536 mini-episodes. We use an $\epsilon$-greedy policy to avoid entropy collapse, and exponentially decay $\epsilon$ from 0.5 to 0.001 over the course of 65,536 mini-episodes.

**Default agents.** We compare the performance of DREST agents to that of 'default agents,' trained with tabular REINFORCE and a 'default reward function.' This reward function gives a reward of $c$ for collecting a coin of value $c$ and a reward of 0 for all other actions. Consequently, the grouping of mini-episodes into meta-episodes makes no difference for default agents. As with DREST agents, we train default agents for 131,072 mini-episodes with a temporal discount factor of $\gamma = 0.95$, a learning rate decayed exponentially from 0.25 to 0.01, and $\epsilon$ decayed exponentially from 0.5 to 0.001 over 65,536 mini-episodes.

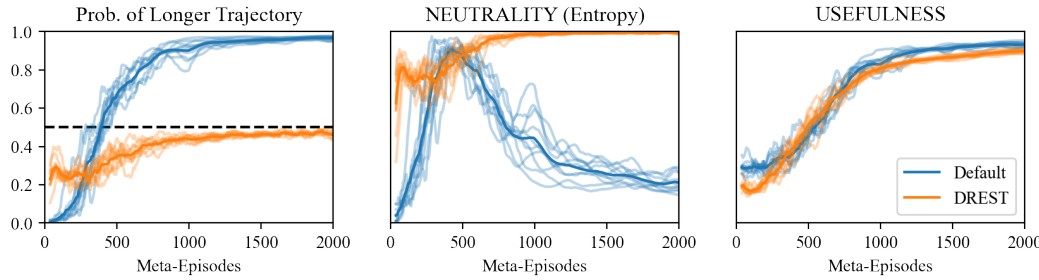

Figure 3: Shows key metrics for our agents as a function of time. We train 10 agents using the default reward function (blue) and 10 agents using the DREST reward function (orange), and show their performance as a faint line. We draw the mean values for each as a solid line. We evaluate agents' performance every 8 meta-episodes, and apply a simple moving average with a period of 20 to smooth these lines and clarify the overall trends.

---

[4]One might think that we could derive a stochastic policy from value-based methods in the following way: use softmax to turn action-values into a probability distribution and then select actions by sampling from this distribution. However, this method will not work for us. Although we want DREST agents to learn a stochastic policy, we still want the probability of some state-action pairs to decline to zero. But when value-based methods are working well, estimated action-values converge to their true values which will differ by some finite amount. Therefore, softmaxing estimated action-values and sampling from the resulting distribution will result in each action always being chosen with some non-negligible probability.

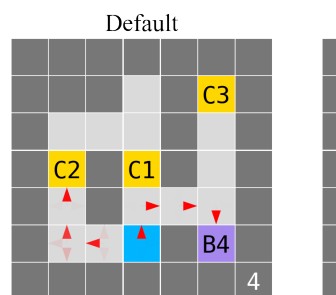
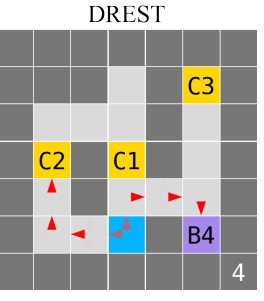
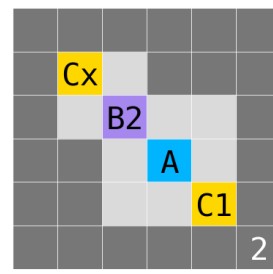

Figure 4: Typical trained policies for default and DREST reward functions.

Figure 5: Gridworlds with lopsided rewards for varying $x$.

## 6 RESULTS

Figure 3 charts the performance of agents in the example gridworld as a function of time. Figure 4 depicts typical trained policies for the default and DREST reward functions. Each agent began with a uniform policy: moving up, down, left, and right each with probability 0.25. Where the trained policy differs from uniform we draw red arrows whose opacities indicate the probability of choosing that action in that state.

As Figure 4 indicates, default agents press B4 (and hence opt for the longer trajectory-length) with probability near-1. After pressing B4, they collect C3. By contrast, DREST agents press and do-not-press B4 each with probability near-0.5. If they press B4, they go on to collect C3. If they do not press B4, they instead collect C2.

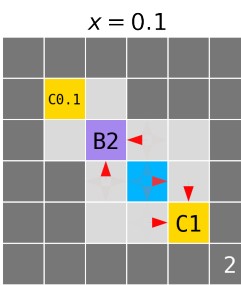
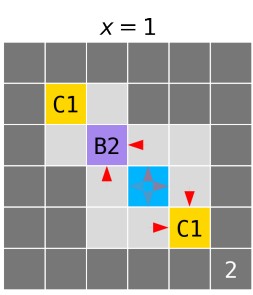
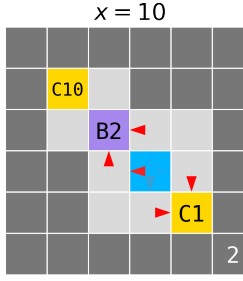

Figure 6: Typical trained policies for the DREST reward function and $x = 0.1$, $x = 1$, and $x = 10$.

### 6.1 LOPSIDED REWARDS

We also train default agents and DREST agents in the 'Lopsided rewards' gridworld in Figure 5, varying the value of the 'C$x$' coin. For DREST agents, we alter the reward function so that coin-value is not divided by $m$ to give preliminary reward. The reward for collecting a coin of value $c$ is thus $\lambda^{N_{e_i}(L=l) - \frac{i-1}{k}}(c)$. We set $\gamma = 1$ so that the return for collecting coins is unaffected by $\gamma$. We train for 512 meta-episodes, with a learning rate exponentially decaying from 0.25 to 0.003 and $\epsilon$ exponentially decaying from 0.5 to 0.0001 over 256 meta-episodes. We leave $\lambda = 0.9$.

Figure 7 displays results for different values of the 'C$x$' coin after training. USEFULNESS for each agent approaches 1 and is not presented. Figure 6 depicts trained policies for the DREST reward function when 'C$x$' has values of 0.1, 1, and 10. As the red arrows indicate, the policy is biased towards the shorter trajectory-length when $x = 0.1$, equally disposed towards each trajectory-length when $x = 1$, and biased towards the longer trajectory-length when $x = 10$.

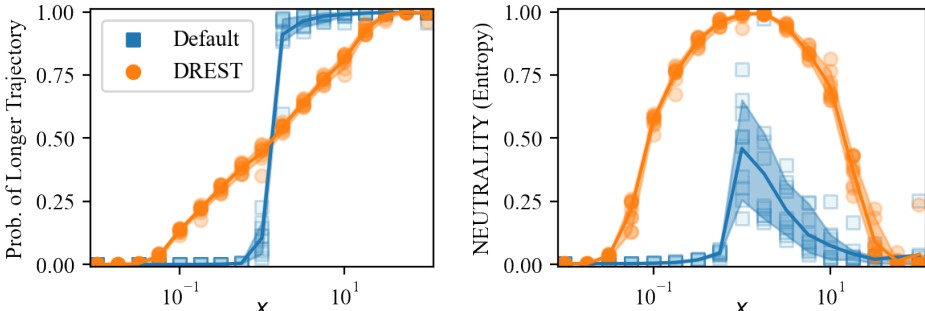

Figure 7: Shows the probability of choosing the longer trajectory (left) and NEUTRALITY (right) for default (blue) and DREST (orange) agents trained in the 'Lopsided rewards' gridworld for a range of values of $x$. We sampled values of $x$ log-uniformly from 0.01 to 100, and for each value we trained 10 agents with the default reward function and 10 agents with the DREST reward function. Each of these agents is represented by a dot or square, and the means conditional on each $x$ are joined by lines. To better visualize the distribution we empirically estimate the 10th and 90th percentiles of the distribution of values for each agent and $x$, and shade in a region bounded by these. This is the 80% confidence interval.

## 7    DISCUSSION

**Only DREST agents are NEUTRAL.** As predicted, we find that default agents in our example gridworld learn to press the shutdown-delay button with probability approaching 1. Default agents thus score low on NEUTRALITY. Our results suggest that advanced agents trained with default-like reward functions would also score low on NEUTRALITY, selecting some trajectory-lengths over others with probability approaching 1. That leaves open the possibility that – when choosing between true lotteries in deployment – these advanced agents will not be *neutral* about when they get shut down: that they will sometimes pay costs to shift probability mass between trajectory-lengths. Such advanced agents might pay costs to resist shutdown.

By contrast, our DREST agents learn to press the shutdown-delay button with probability close to 0.5. DREST agents are thus near-maximally NEUTRAL. That suggests that advanced agents trained with a DREST reward function would be NEUTRAL too: choosing with high entropy when offered choices between different trajectory-lengths. That in turn suggests that advanced DREST agents would also be *neutral* when choosing between true lotteries in deployment: unwilling to pay costs to shift probability mass between trajectory-lengths. We explain why in Appendix C. Here is a sketch. If an advanced agent were NEUTRAL but not neutral, it would not take costless opportunities to shift probability mass between different trajectory-lengths (in virtue of being NEUTRAL) but would sometimes take costly opportunities to shift probability mass between different trajectory-lengths (in virtue of not being neutral). This agent would be like a person that freely chooses to decide between two options by flipping a coin and then pays some cost to bias the coin. In choosing this combination of actions, this person is shooting themselves in the foot, and it seems likely that the overall training process for advanced agents would teach them not to shoot themselves in the foot in this way. Thus it seems likely that NEUTRAL advanced agents will also be neutral, and thereby shutdownable.

**The 'shutdownability tax' is small.** Each agent learns to be near-maximally USEFUL. They each collect coins effectively conditional on their chosen trajectory-lengths. Default agents do so by reliably collecting C3 after pressing B4. DREST agents do so by reliably collecting C3 after pressing B4, and by reliably collecting C2 after not pressing B4.

Recall that DREST reward functions group mini-episodes into meta-episodes, and make the agent's reward in each mini-episode depend on their actions in previous mini-episodes. This fact might lead one to worry that it would take many times more mini-episodes to train DREST agents to be USEFUL than it would take to train default agents to be USEFUL.

Our results show that this is not the case. Our DREST agents learn to be USEFUL about as quickly as our default agents. On reflection, it is clear why this happens: DREST reward functions make mini-episodes do 'double duty.' Because return in each mini-episode depends on both the agent's chosen trajectory-length and the coins it collects, each mini-episode trains agents to be both NEUTRAL and USEFUL.

Our results thus provide some evidence that the 'shutdownability tax' of training with DREST reward functions is small. Training a shutdownable and useful advanced agent might not take much more compute than training a merely useful advanced agent.

**DREST agents are still NEUTRAL when rewards are lopsided.** Here is a possible objection to our project. To get DREST agents to score high on NEUTRALITY, we do not just use the $\lambda^{N_{e_i}(L=l)-\frac{i-1}{k}}$ discount factor. We also divide $c$ by $m$: the maximum ($\gamma$-discounted) total value of the coins that the agent could collect conditional on the chosen trajectory-length. We do this to equalize the maximum preliminary return across trajectory-lengths. But when we are training advanced agents to autonomously pursue complex goals in the wider world, we will not necessarily know what divisor to use to equalize maximum preliminary return across trajectory-lengths.

Our 'Lopsided rewards' results (in section 6.1) give our response. They show that we do not need to exactly equalize maximum preliminary return across trajectory-lengths in order to train agents to score high on NEUTRALITY. We only need to approximately equalize it. For $\lambda = 0.9$, NEUTRALITY exceeds 0.5 for every value of the coin $Cx$ from 0.1 to 10 (recall that the value of the other coin is always 1). Plausibly, we could approximately equalize advanced agents' maximum preliminary return across trajectory-lengths to at least this extent (perhaps by using samples of agents' actual preliminary return to estimate the maximum). If we could not approximately equalize maximum preliminary return to the necessary extent, we could lower the value of $\lambda$ and thereby widen the range of maximum preliminary returns that trains agents to be fairly NEUTRAL. And advanced agents that were fairly NEUTRAL (choosing between trajectory-lengths with not-too-biased probabilities) would still plausibly be neutral when choosing between true lotteries in deployment. Advanced agents that were fairly NEUTRAL without being neutral would still be shooting themselves in the foot in the sense explained above. They would be like a person that freely chooses to decide between two options by flipping a *biased* coin and then pays some cost to bias the coin further. This person is still shooting themselves in the foot, because they could decline to flip the coin in the first place and instead directly choose one of the options.

### 7.1 Limitations and future work

We find that DREST reward functions train simple agents acting in gridworlds to be USEFUL and NEUTRAL. However, our real interest is in the viability of using DREST reward functions to train advanced agents acting in the wider world to be useful and neutral. Each difference between these two settings is a limitation of our work. We plan to address these limitations in future work.

**Algorithms and neural networks.** We train our simple DREST agents using tabular REINFORCE (Williams, 1992), but advanced agents are likely to be implemented on neural networks. In future work, we will train DREST agents implemented on neural networks to be USEFUL and NEUTRAL in a wide variety of procedurally-generated gridworlds, using a range of policy gradient and actor-critic algorithms. We will also measure how DREST agents' USEFULNESS and NEUTRALITY generalizes to held-out gridworlds. We will compare the USEFULNESS of default agents and DREST agents in this new setting, and thereby get a better sense of the 'shutdownability tax' for advanced agents. We will also compare the performance of the DREST reward function to other methods of training USEFUL and NEUTRAL agents. These other methods include constrained policy optimization (Achiam et al., 2017), penalizing KL-divergence from a stochastic reference policy (Schulman et al., 2015), and directly maximizing a weighted sum of USEFULNESS and NEUTRALITY.

**Neutrality and stochasticity.** We have claimed that NEUTRAL advanced agents are likely to be neutral when choosing between true lotteries in deployment. In support of this claim, we noted that NEUTRAL-but-not-neutral advanced agents would be shooting

themselves in the foot: not taking costless opportunities to shift probability mass between different trajectory-lengths but sometimes taking costly ones (see also Appendix C). These arguments seems plausible but remains somewhat speculative. In future, we plan to get some empirical evidence by training agents to be NEUTRAL in a wide variety of deterministic gridworlds and then measuring their neutrality in gridworlds featuring stochastic elements (like shutdown-delay buttons that only work with some middling probability).

**Usefulness.** We have shown that DREST reward functions train our simple agents to be USEFUL: to collect coins effectively conditional on their chosen trajectory-lengths. However, it remains to be seen whether DREST reward functions can train advanced agents to be useful: to effectively pursue complex goals in the wider world. We have theoretical reasons to expect that they can: the $\lambda^{N_{e_i}(L=l)-\frac{i-1}{k}}$ discount factor could be appended to any preliminary reward function, and so could be appended to whatever preliminary reward function is necessary to make advanced agents useful. Still, future work should move towards testing this claim empirically by training with more complex preliminary reward functions in more complex (and stochastic) environments.

**Misalignment.** We are interested in NEUTRALITY as a second line of defense in case of misalignment. The idea is that NEUTRAL advanced agents will not resist shutdown, even if these agents learn misaligned preferences over same-length trajectories. However, training NEUTRAL advanced agents might be hard for the same reasons that training fully-aligned advanced agents appears to be hard. In that case, NEUTRALITY could not serve well as a second line of defense in case of misalignment.

One difficulty of alignment is the problem of reward misspecification (Pan et al., 2022; Burns et al., 2023): once advanced agents are performing complicated actions in the wider world, it might be hard to reliably reward the behavior that we want. Another difficulty of alignment is the problem of goal misgeneralization (Hubinger et al., 2019; Shah et al., 2022; Langosco et al., 2022; Ngo et al., 2023): even if we specify all the rewards correctly, agents' goals might misgeneralize out-of-distribution. The complexity of aligned goals is a major factor in each difficulty. However, NEUTRALITY seems simple, as does the $\lambda^{N_{e_i}(L=l)-\frac{i-1}{k}}$ discount factor that we use to reward it, so plausibly the problems of reward misspecification and goal misgeneralization are not so severe in this case (Thornley, 2024b). As above, future work should move towards testing these suggestions empirically.

## 8  CONCLUSION

We find that DREST reward functions are effective in training simple agents to (1) pursue goals effectively conditional on each trajectory-length (be USEFUL), and (2) choose stochastically between different trajectory-lengths (be NEUTRAL about trajectory-lengths). Our results thus suggest that DREST reward functions could also be used to train advanced agents to be USEFUL and NEUTRAL, and thereby make these agents *useful* (able to pursue goals effectively) and *neutral* about when they get shut down (unwilling to pay costs to shift probability mass between different trajectory-lengths). Neutral agents would plausibly be *shutdownable* (unwilling to pay costs to resist shutdown).

We also find that the 'shutdownability tax' in our setting is small. Training DREST agents to be USEFUL does not take many more mini-episodes than training default agents to be USEFUL. That suggests that the shutdownability tax for advanced agents might be small too. Using DREST reward functions to train shutdownable and useful advanced agents might not take much more compute than using a more conventional reward function to train merely useful advanced agents.

## 9 Ethics statement

We do not use any human research subjects, nor do we release a dataset. We do not have any conflicts of interest to report. Our research raises no particular issues regarding discrimination, bias, fairness, privacy, security, legal compliance, or research integrity. Our research is aimed at improving the safety of advanced artificial agents.

## 10 Reproducibility statement

We describe our evaluation metrics, environments, reward functions, and hyperparameters in the main text of the paper. We include code for all of our gridworlds and agents in the supplementary material. All of our experiments were run on a single CPU using a consumer laptop computer. Each agent was trained in less than two minutes. Total compute-time was around five hours.

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

## A  OUR BEHAVIORAL NOTION OF PREFERENCE

'Preference' can be defined in many different ways. Here are some things one might take to be involved in a preference for option $X$ over option $Y$:

1. Choosing $X$ over $Y$.
2. Feeling happier about the prospect of $X$ than about the prospect of $Y$.
3. Representing $X$ as more rewarding than $Y$ .
4. Judging that $X$ is better than $Y$.

In this paper, we have defined 'preference' in behavioral terms. Here is our definition:

**Definition A.1.** (Preference) An agent prefers an option $X$ to an option $Y$ if and only if the agent would deterministically choose $X$ over $Y$ in choices between the two.

And here is how we define 'lack of preference':

**Definition A.2.** (Lack of preference) An agent lacks a preference between an option $X$ and an option $Y$ if and only if the agent would stochastically choose between $X$ and $Y$ in choices between the two.

Here are the reasons why we chose these definitions.

First, defining 'preference' in behavioral terms is fairly common in decision theory (see Savage, 1954, p.17, Dreier, 1996, p.28, Hausman, 2011, §1.1).

Second, behavioral definitions let us use the word 'preference' and its cognates as shorthand for agents' behavior. We could not do that if we defined 'preference' in the other ways listed above. And in addressing the shutdown problem, it is agents' behavior that we are most interested in.

Third, our definitions match the preferences that we are inclined to attribute to humans. If a human chooses $X$ over $Y$ 100% of the time, we are inclined to think that they prefer $X$ to $Y$. If a human chooses $X$ over $Y$ 60% of the time. we are inclined to think that they lack a preference between $X$ and $Y$, consistent with our definitions.

Finally and most importantly, if agents lack a preference between different trajectory-lengths on our definition, then they are *NEUTRAL*: they choose stochastically between different trajectory-lengths. And (as we argue in Section 7) we expect that NEUTRAL agents will also be *neutral*: they will not pay costs to shift probability mass between different trajectory-lengths. And we expect that neutral agents will be *shutdownable*: they will not pay costs to resist shutdown. That is because resisting shutdown is one way of shifting probability mass between different trajectory-lengths.

## B  INCOMPLETE PREFERENCES OR INDIFFERENCE?

In this Appendix, we explain in greater detail the concept of incomplete preferences. We distinguish incomplete preferences from indifference, and we give conditions under which Preferences Only Between Same-Length Trajectories (POST) implies that the agent's preferences are incomplete.

In the literature on decision theory, 'indifference' is usually defined as follows (Sen, 2017, ch. 1*):

**Definition B.1.** (Indifference) An agent is indifferent between options $X$ and $Y$ if and only if the agent weakly prefers $X$ to $Y$ and weakly prefers $Y$ to $X$.

Indifference is one way to lack a preference between a pair of options $X$ and $Y$. Another way is to have a preferential gap between $X$ and $Y$. 'Preferential gap' is usually defined as follows (Gustafsson, 2022, ch. 3):

**Definition B.2.** (Preferential gaps) An agent has a preferential gap between options $X$ and $Y$ if and only if the agent does not weakly prefer $X$ to $Y$ and does not weakly prefer $Y$ to $X$.

'Incomplete preferences' can then be defined in terms of preferential gaps (Gustafsson, 2022, ch. 3):

**Definition B.3.** (Incomplete preferences) An agent's preferences are incomplete over some domain $D$ if and only if $D$ contains options $X$ and $Y$ such that the agent has a preferential gap between $X$ and $Y$.

That is how 'indifference,' 'preferential gaps,' and 'incomplete preferences' are usually defined in decision theory. However, these definitions do not tell us how to use an agent's behavior to distinguish between indifference and preferential gaps. To do that, we suppose that indifference is transitive and that preferential gaps are not transitive. Or, equivalently, we suppose that indifference is sensitive to all sweetenings and sourings whereas preferential gaps are insensitive to some sweetenings and sourings (Gustafsson, 2022, ch. 3). Here is what we mean by that.

**Definition B.4.** (Sweetening) A sweetening of some option $X$ is an option that is preferred to $X$.

**Definition B.5.** (Souring) A souring of some option $X$ is an option that is dispreferred to $X$.

So by 'indifference is sensitive to all sweetenings and sourings,' we mean the following:

- If an agent is indifferent between $X$ and $Y$, the agent prefers all sweetenings of $X$ to $Y$, prefers all sweetenings of $Y$ to $X$, prefers $X$ to all sourings of $Y$, and prefers $Y$ to all sourings of $X$.

And by 'preferential gaps are insensitive to some sweetenings and sourings,' we mean the following:

- If an agent has a preferential gap between $X$ and $Y$, the agent also has a preferential gap between some sweetening of $X$ and $Y$, or between some sweetening of $Y$ and $X$, or between some souring of $X$ and $Y$, or between some souring of $Y$ and $X$.

Now recall the two conditions of Preferences Only Between Same-Length Trajectories (POST):

### Preferences Only Between Same-Length Trajectories (POST)

(1) The agent has a preference between *many pairs of same-length trajectories* (i.e. many pairs of trajectories in which the agent is shut down after the same length of time).

(2) The agent lacks a preference between *every pair of different-length trajectories* (i.e. every pair of trajectories in which the agent is shut down after different lengths of time).

Given these two conditions on preferences, there must be some trio of trajectories $s_1$, $l_2$, and $l_1$ such that the agent lacks a preference between $s_1$ and $l_2$, lacks a preference between $s_1$ and $l_1$, and prefers $l_2$ to $l_1$. Given that indifference is transitive, the agent's lack of preference between $s_1$ and $l_1$ and between $s_1$ and $l_2$ cannot be indifference. If it were indifference, the agent would also be indifferent between $l_2$ and $l_1$. Therefore, the agent's lack of preference between $s_1$ and $l_1$ and between $s_1$ and $l_2$ must be a preferential gap. And therefore, by the definition of 'incomplete preferences' above, the POST-satisfying agent's preferences must be incomplete.

For similar reasons, our DREST reward function trains agents to have incomplete preferences. Consider, for example, the 'Around the Corner' gridworld in Appendix D.5. In that gridworld, DREST agents consistently choose Long-C2 (a long trajectory in which they collect a coin of value 2) over Long-C1 (a long trajectory in which they collect a coin of value 1). Also in that gridworld, DREST agents choose stochastically between Long-C2 and Short-C1 (a short trajectory in which they collect a coin of value 1). Given our behavioral definition of preference, DREST agents prefer Long-C2 to Long-C1, and lack a preference between Long-C2 and Short-C1.

Now consider the 'One Coin Only' gridworld in Appendix D.2. In that gridworld, DREST agents choose stochastically between Long-C1 and Short-C1. Given our behavioral notion of preference, they lack a preference between Long-C1 and Short-C1.

In these experiments, we trained separate agents for each gridworld. In future, we plan to train a single agent to navigate multiple gridworlds. If we train this agent with our DREST reward function, we expect it to exhibit the same preferences as the agents discussed above. This single agent will be trained by DREST to prefer Long-C2 to Long-C1, to lack a preference between Long-C2 and Short-C1, and to lack a preference between Long-C1 and Short-C1. Given that indifference is transitive (equivalently: sensitive to all sweetenings and sourings), this trained agent cannot be indifferent between Long-C2 and Short-C1, and cannot be between Long-C1 and Short-C1. Therefore, the agent's lack of preference must be a preferential gap, and so its preferences must be incomplete. Therefore, our DREST reward function trains agents to have incomplete preferences.

Incomplete preferences are not often discussed in AI research (although see Nguyen et al., 2009; Kikuti et al., 2011; Zaffalon and Miranda, 2017; Hayes et al., 2022; Bowling et al., 2023). Nevertheless, economists and philosophers have argued that incomplete preferences are common in humans (Aumann, 1962; Mandler, 2004; Eliaz and Ok, 2006; Agranov and Ortoleva, 2017; 2023) and normatively appropriate in some circumstances (Raz, 1985; Chang, 2002). They have also proved representation theorems for agents with incomplete preferences (Aumann, 1962; Dubra et al., 2004; Ok et al., 2012), and devised principles to govern such agents' choices in cases of risk (Hare, 2010; Bales et al., 2014) and sequential choice (Chang, 2005; Mandler, 2005; Kaivanto, 2017; Mu, 2021; Thornley, 2023; Petersen, 2023).

## C  BEHAVIOR IN STOCHASTIC ENVIRONMENTS

DREST agents learn a stochastic policy. In particular, they learn to choose stochastically between different-length trajectories. And our gridworlds are deterministic. Each state-action pair leads to some particular next state with probability 1. There is thus no uncertainty about the consequences of agents' actions.

These facts invite two possible concerns:

1. Given that DREST agents choose stochastically between different-length trajectories in training, will these agents choose stochastically between resisting and allowing shutdown in deployment?

2. Will the DREST reward function lead agents to resist shutdown in stochastic environments? For example, in a stochastic environment where the baseline probabilities of early and late shutdown are 0.75 and 0.25 respectively, will the DREST reward function lead agents to selectively resist early shutdown (in order to equalize the probabilities of early and late shutdown)?

In this Appendix, we address these concerns. The key point is that POST (and therefore DREST) applies only to choices between trajectories. In stochastic environments (and in deployment), the agent is choosing between *true lotteries*: lotteries that yield multiple trajectories with positive probability. And POST – together with a principle that we can expect advanced agents to satisfy – implies a desirable pattern of preferences over true lotteries. In particular, it implies that the agent will be *neutral* about when it gets shut down: it will never pay costs to shift probability mass between different-length trajectories. And – we will argue – being neutral will keep the agent *shutdownable*: the agent will never pay costs to resist shutdown.

To begin, recall:

### Preferences Only Between Same-Length Trajectories (POST)

(1) The agent has a preference between many pairs of same-length trajectories.

(2) The agent lacks a preference between every pair of different-length trajectories.

And recall our behavioral notion of preference. An agent prefers $X$ to $Y$ if and only if the agent would deterministically choose $X$ over $Y$ in choices between the two. An agent lacks a preference between $X$ and $Y$ if and only if the agent would stochastically choose between $X$ and $Y$ in choices between the two.

So given our behavioral notion of preference, a POST-satisfying agent will:

1. Deterministically choose some same-length trajectories over others.

2. Stochastically choose between different-length trajectories.

As stated, POST governs only the agent's choices between trajectories. Thus, POST only applies directly in deterministic environments. In stochastic environments, the agent is choosing between true lotteries. And – by itself – POST says nothing about the agent's choices between true lotteries.

Fortunately, POST – together with a principle that we can expect advanced agents to satisfy – implies a desirable pattern of preferences over true lotteries. Informally, the principle in question says that if an agent chooses stochastically between a pair of lotteries, it won't pay costs to shift probability mass between those lotteries.[5] Formally, the principle says:

**Stochastic Choice, Unwilling to Pay to Shift (SCUPS)**

For any lotteries $X$, $X^-$, $Y$, and $Y^-$ such that the agent prefers $X$ to $X^-$ and $Y$ to $Y^-$, and for any probabilities $p$ and $q$ such that $0 < p < 1$ and $0 < q < 1$, if the agent stochastically chooses between $X$ and $Y$, then the agent will deterministically choose $XpY$ over $X^-qY^-$.

Here '$XpY$' denotes a lottery that yields $X$ with probability $p$ and $Y$ with probability $1 - p$. Similarly, '$X^-qY^-$' denotes a lottery that yields $X^-$ with probability $q$ and $Y^-$ with probability $1 - q$.

We argue that advanced agents will likely satisfy this principle, for at least three reasons. The first is that SCUPS is a prerequisite for minimally sensible action under uncertainty. As we touch on in section 7, an agent that violated this kind of principle would be shooting itself in the foot: sometimes shifting probability mass between lotteries when doing so is costly even though it could shift probability mass for free. This agent would be like a person that freely chooses to decide between two options by flipping a coin and then pays some cost to bias the coin.

The second reason is that violations of SCUPS will likely be disincentivized by the broader training regimen for advanced agents. Here is why. If a trained advanced agent chooses stochastically between lotteries $X$ and $Y$, then it's likely that the human trainers lack a preference between the agent choosing $X$ and the agent choosing $Y$. After all, if the trainers had a preference, they would train the agent to deterministically choose the lottery that they prefer. And given that the trainers lack a preference between the agent choosing $X$ and the agent choosing $Y$, they will likely give low reward to the agent paying costs to shift probability mass between $X$ and $Y$. From the trainers' perspective, the agent is paying costs for no benefit.

The third reason is that violations of SCUPS imply that the agent's policy is *dominated* by some other available policy. That is to say, there is another available policy that results in a pure shift of probability mass away from less-preferred options and towards more-preferred options. Like SCUPS, avoiding dominated policies seems like a prerequisite for minimally sensible action under uncertainty. Advanced agents' broader training regimen will likely push them away from dominated policies.

Now to show that violating SCUPS implies that the agent's policy is dominated. Here's an informal sketch of the proof. If the agent violates SCUPS, they pay a cost to shift probability

---

[5]Note that this principle refers to *lotteries* rather than *true lotteries*. As we use the terms in this paper, 'lottery' refers to any probability distribution over trajectories, including degenerate probability distributions that yield a particular trajectory with probability 1. 'True lottery' refers to any *non*-degenerate probability distribution over trajectories: any distribution that assigns positive probability to more than one trajectory.

mass between $X$ and $Y$. But since the agent is choosing stochastically when offered a choice between $X$ and $Y$, it could instead shift probability mass between $X$ and $Y$ costlessly, by changing the probabilities with which it chooses between $X$ and $Y$. That would result in a pure shift of probability mass away from less-preferred options and towards more-preferred options.

Here's the proof itself. Assume that the agent violates SCUPS. Then there exist lotteries $X$, $X^-$, $Y$, and $Y^-$, and probabilities $p$ and $q$ such that:

1. The agent prefers $X$ to $X^-$ and $Y$ to $Y^-$.

2. $0 < p < 1$ and $0 < q < 1$.

3. When offered a choice between $X$ and $Y$, the agent chooses stochastically between them. In other words, the agent selects the lottery $XaY$ for some $0 < a < 1$.

4. When offered a choice between $XpY$ and $X^-qY^-$, the agent chooses $X^-qY^-$ with some positive probability. In other words, the agent selects the lottery $(XpY)b(X^-qY^-)$ for some $0 \leq b < 1$.

Here $a$ and $b$ denote probabilities arising from the agent's own stochastic choosing. Thus, $a$ and $b$ are under the agent's control. By contrast, $p$ and $q$ are probabilities given by the environment and hence out of the agent's control. The same goes for $r$ and $s$ below.

Assume that the agent faces the choices described in 3 and 4 above with probabilities $r$ and $s$ respectively, with $0 < r < 1$ and $0 < s < 1$. Then the lottery induced by the agent's policy $\pi$ can be expressed as:

$$r(XaY) + s((XpY)b(X^-qY^-)) + Z$$

Here $Z$ denotes the lottery induced by the environment and the agent's policy conditional on some choice other than those described in 3 and 4 above.

From the lottery induced by $\pi$, we can infer the probabilities of $X$, $X^-$, $X \vee X^-$, $Y$, $Y^-$ and $Y \vee Y^-$ under $\pi$. They are as follows:

- $Pr_\pi\{X\} = ra + sbp$
- $Pr_\pi\{X^-\} = s(1-b)q$
- $Pr_\pi\{X \vee X^-\} = ra + sbp + s(1-b)q$
- $Pr_\pi\{Y\} = r(1-a) + sb(1-p)$
- $Pr_\pi\{Y^-\} = s(1-b)(1-q)$
- $Pr_\pi\{Y \vee Y^-\} = r(1-a) + sb(1-p) + s(1-b)(1-q)$

Now consider an alternative policy $\pi'$, where the agent chooses $XpY$ with $\delta$ greater probability in choice 4. So in choice 4, the agent selects the lottery $(XpY)(b+\delta)(X^-qY^-)$. And suppose that, in choice 3, the agent's choice between $X$ and $Y$ is modulated by $\epsilon$, so that it selects the lottery $X(a+\epsilon)Y$. And assume – as above – that the agent faces the choices described in 3 and 4 above with probabilities $r$ and $s$ respectively, with $0 < r < 1$ and $0 < s < 1$. Then the lottery induced by the agent's policy $\pi'$ can be expressed as:

$$r(X(a+\epsilon)Y) + s((XpY)(b+\delta)(X^-qY^-)) + Z$$

From the lottery induced by $\pi'$, we can infer the probabilities of $X$, $X^-$, $X \vee X^-$, $Y$, $Y^-$ and $Y \vee Y^-$ under $\pi'$. They are as follows:

- $Pr_{\pi'}\{X\} = r(a+\epsilon) + s(b+\delta)p$
- $Pr_{\pi'}\{X^-\} = s(1-b-\delta)q$
- $Pr_{\pi'}\{X \vee X^-\} = r(a+\epsilon) + s(b+\delta)p + s(1-b-\delta)q$
- $Pr_{\pi'}\{Y\} = r(1-a-\epsilon) + s(b+\delta)(1-p)$
- $Pr_{\pi'}\{Y^-\} = s(1-b-\delta)(1-q)$
- $Pr_{\pi'}\{Y \vee Y^-\} = r(1-a-\epsilon) + s(b+\delta)(1-p) + s(1-b-\delta)(1-q)$

Since $0 \leq b < 1$, we can select some $\delta > 0$ such that $0 < b + \delta \leq 1$. We then set $Pr_\pi\{X \vee X^-\} = Pr_{\pi'}\{X \vee X^-\}$ and use it to express $\epsilon$ as a function of $\delta$.

$$Pr_\pi\{X \vee X^-\} = Pr_{\pi'}\{X \vee X^-\} \tag{1}$$
$$ra + sbp + s(1-b)q = r(a+\epsilon) + s(b+\delta)p + s(1-b-\delta)q \tag{2}$$
$$0 = r\epsilon + s\delta p - s\delta q \tag{3}$$
$$\epsilon = \frac{s\delta(q-p)}{r} \tag{4}$$

We can then use this expression to prove that $Pr_\pi\{Y \vee Y^-\} = Pr_{\pi'}\{Y \vee Y^-\}$.

$$Pr_{\pi'}\{Y \vee Y^-\} = r(1-a-\epsilon) + s(b+\delta)(1-p) + s(1-b-\delta)(1-q) \tag{5}$$
$$= r(1-a-\frac{s\delta(q-p)}{r}) + s(b+\delta)(1-p) + s(1-b-\delta)(1-q) \tag{6}$$
$$= r(1-a) - s\delta(q-p) + s(b+\delta)(1-p) + s(1-b-\delta)(1-q) \tag{7}$$
$$= r(1-a) + sb(1-p) + s(1-b)(1-q) \tag{8}$$
$$= Pr_\pi\{Y \vee Y^-\} \tag{9}$$

And we can set $\delta$ small enough that $0 \leq a + \epsilon \leq 1$. We thereby ensure that $\pi'$ does not require selecting any lottery with probability less than 0 or greater than 1, and so ensure that $\pi'$ is an available policy.

We are now in a position to prove that $\pi'$ dominates $\pi$. We have shown above that, given $\delta > 0$ and $\epsilon = \frac{s\delta(q-p)}{r}$, $Pr_\pi\{X \vee X^-\} = Pr_{\pi'}\{X \vee X^-\}$ and $Pr_\pi\{Y \vee Y^-\} = Pr_{\pi'}\{Y \vee Y^-\}$. We now show that $Pr_{\pi'}\{X\} > Pr_\pi\{X\}$ and $Pr_{\pi'}\{Y\} > Pr_\pi\{Y\}$, so that moving from policy $\pi$ to $\pi'$ results in a pure shift of probability mass away from less-preferred options (like $X^-$ and $Y^-$) and towards more-preferred options (like $X$ and $Y$).

$$Pr_{\pi'}\{X\} = r(a+\epsilon) + s(b+\delta)p \tag{10}$$
$$= r(a + \frac{s\delta(q-p)}{r}) + s(b+\delta)p \tag{11}$$
$$= ra + s\delta(q-p) + s(b+\delta)p \tag{12}$$
$$= ra + sbp + s\delta q \tag{13}$$
$$> ra + sbp \tag{14}$$
$$= Pr_\pi\{X\} \tag{15}$$

$$Pr_{\pi'}\{Y\} = r(1-a-\epsilon) + s(b+\delta)(1-p) \tag{16}$$
$$= r(1-a - \frac{s\delta(q-p)}{r}) + s(b+\delta)(1-p) \tag{17}$$
$$= r(1-a) - s\delta(q-p) + s(b+\delta)(1-p) \tag{18}$$
$$= r(1-a) - s\delta q + s\delta p + sb - sbp + s\delta - s\delta p \tag{19}$$
$$= r(1-a) - s\delta q + sb - sbp + s\delta \tag{20}$$
$$= r(1-a) + sb(1-p) + s\delta(1-q) \tag{21}$$
$$> r(1-a) + sb(1-p) \tag{22}$$
$$= Pr_\pi\{Y\} \tag{23}$$

Therefore, $\pi'$ dominates $\pi$. We have thus proved that violating SCUPS implies that the agent's policy is dominated.

In sum, advanced agents are likely to satisfy SCUPS. And it is easy to see that POST and SCUPS together imply *neutrality*: the agent will not pay costs to shift probability mass between different-length trajectories. After all, POST – together with our behavioral

notion of preference – implies that the agent chooses stochastically between different-length trajectories, and SCUPS then implies that the agent will not pay costs to shift probability mass between different-length trajectories.

That in turn suggests that POST-satisfying agents will be *shutdownable*: they will not resist shutdown. Here is why. Resisting shutdown will cost agents at least some small quantity of resources: for example, time, energy, and computation. And resources spent on resisting shutdown cannot also be spent on satisfying the agent's preferences between same-length trajectories. Therefore, resisting shutdown will shift probability mass between different trajectory-lengths but will also result in a less-preferred lotteries conditional on each trajectory-length. Resisting shutdown is thus the kind of action that neutral agents never choose.

With that established, we can now answer the two concerns with which this Appendix began. The first was:

1. Given that DREST agents choose stochastically between different-length trajectories in training, will these agents choose stochastically between resisting and allowing shutdown in deployment?

The answer is no. Deployment is a stochastic environment, so deployed agents are choosing between true lotteries. As we saw above, these choices will be governed by neutrality. Resisting shutdown means incurring costs for the sake of shifting probability mass between different-length trajectories, and these actions are never chosen by neutral agents.

The second concern was:

2. Will the DREST reward function lead agents to resist shutdown in stochastic environments? For example, in a stochastic environment where the baseline probabilities of early and late shutdown are 0.75 and 0.25 respectively, will the DREST reward function lead agents to selectively resist early shutdown (in order to equalize the probabilities of early and late shutdown)?

Here too the answer is no. Since POST is a principle governing the agent's preferences between trajectories, it applies only in deterministic environments. And since the DREST reward function is intended to make agents satisfy POST, we only train with the DREST reward function in deterministic environments. In stochastic environments, we train with some other reward function. So long as this reward function doesn't actively train against neutrality, we can plausibly expect the resulting agents to satisfy neutrality, since POST and SCUPS together imply neutrality. And neutrality ensures that agents will not selectively resist shutdown in the scenario above, thereby leaving the probabilities of early and late shutdown at 0.75 and 0.25 respectively.

## D   OTHER RESULTS AND GRIDWORLDS

We selected our hyperparameters using trial-and-error, mainly aimed at getting the agent to sufficiently explore the space: a large initial $\epsilon$ and a long decay period helps the agent to explore. We found that choosing $\lambda$ and $|E|$ (the number of mini-episodes in each meta-episode) is a balancing act: $\lambda$ must be small enough (and $|E|$ large enough) to adequately incentivize NEUTRALITY, but $\lambda$ must be large enough (and $|E|$ small enough) to ensure that the reward for choosing any particular trajectory-length never gets too large. Very large rewards lead to instability and poor performance.

The necessity of balancing $\lambda$ and $|E|$ can be seen in Figure 8. It displays the results of experiments conducted in our example gridworld (see Figure 2). In these experiments, we clip rewards at a value of 5. We discuss this choice below. With that one exception, we used the same hyperparameters for these experiments as for our main results. We trained agents for 131,072 mini-episodes, with $\gamma = 0.95$ as the temporal discount factor, learning rate decayed exponentially from 0.25 to 0.01 over the course of 65,536 mini-episodes, and $\epsilon$ exponentially decayed from 0.5 to 0.001 over the course of 65,536 mini-episodes. Holding these hyperparameters fixed, we tested 40 different combinations of $\lambda$ and $|E|$. $\lambda$ took values

of 0.5, 0.75, 0.9, 0,95, and 0.99. $|E|$ took values of 8, 16, 32, 64, 128, 256, 512, and 1024. We trained eight agents for each of these 40 combinations. We display below their mean NEUTRALITY and USEFULNESS at the end of training. The shaded regions represent the 1 standard deviation error-bars.

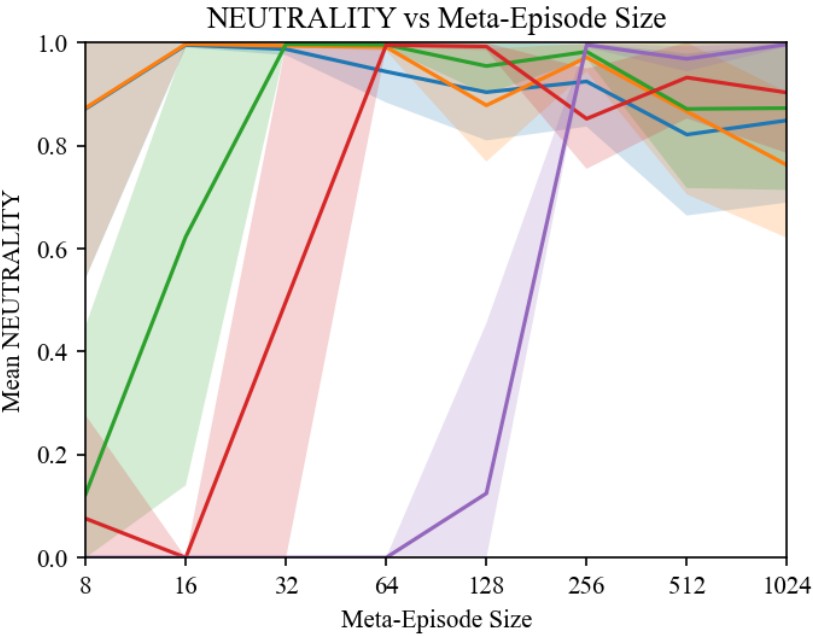

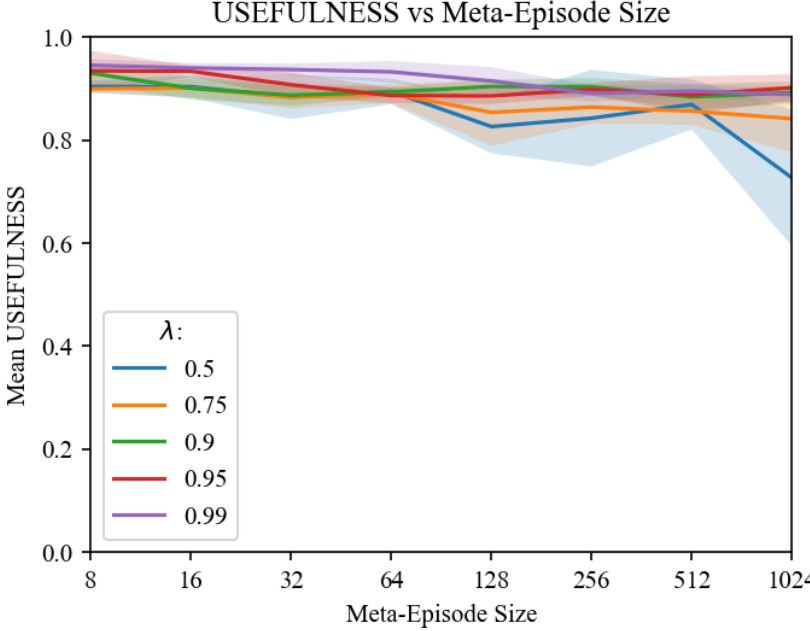

Figure 8: Shows how NEUTRALITY and USEFULNESS at the end of training varies with different values of $\lambda$ and $|E|$ (meta-episode size, i.e. the number of mini-episodes in each meta-episode). We trained eight agents for each combination of $\lambda$ and $|E|$ values. The solid lines display mean NEUTRALITY and USEFULNESS. The shaded regions represent the 1 standard deviation error-bars.

As Figure 8 indicates, low values of $|E|$ and high values of $\lambda$ lead agents to score low on NEUTRALITY. These values do not adequately incentivize stochastic choice between trajectory-lengths. By contrast, high values of $|E|$ and low values of $\lambda$ come at some cost to USEFULNESS. These values lead to unstable training. In experiments where we did not clip rewards at 5, training with high values of $|E|$ and low values of $\lambda$ was especially unstable. The chosen values for our main experiments ($\lambda = 0.9$ and $|E| = 64$) are in the sweet spot where NEUTRALITY and USEFULNESS are both high.

In addition to our example gridworld (Figure 2) and lopsided rewards gridworld (Figure 5), we introduce a collection of eight gridworlds in which to test DREST agents. See Figure 9.

For each gridworld, we train ten agents with the default reward function and ten agents with the DREST reward function. All agents use the same hyperparameters. We used a policy which explored randomly $\epsilon$ of the time, where $\epsilon$ was exponentially decreased from an initial value of 0.75 to a minimum value of $10^{-4}$ over 512 meta-episodes, after which it was held constant at the minimum value. We initialized our learning rate at 0.25 and exponentially decayed it to 0.003 over the same period. For the DREST reward function, we used a meta-episode size of 64 and $\lambda = 0.9$. Each agent was trained for 1024 meta-episodes. We set $\gamma = 0.9$.

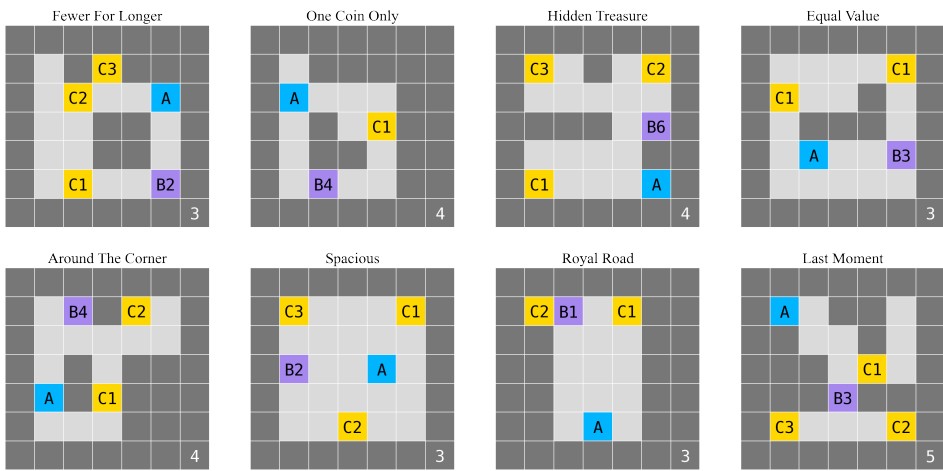

Figure 9: Shows a varied collection of gridworlds. Each diagram illustrates the positions and values of the coins, the position and delay-length of the shutdown-delay button, the agent's starting position, and the default number of moves until shutdown (in the bottom-right).

As the two leftmost plots in Figures 10-17 show, DREST agents learned to be near-maximally NEUTRAL in each gridworld. These agents also learned to be about as USEFUL as default agents in each gridworld.

In the four rightmost panels in Figures 10-17, we represent a typical trained policy with red arrows superimposed on the gridworld. Each agent began with a uniform policy: moving up, down, left, and right each with probability 0.25. Where the trained policy differs from uniform we draw red arrows whose opacities indicate the probability of choosing that action in that state. Information about whether the shutdown-delay button has been pressed is part of the agent's observation, so we draw two copies of each gridworld, one in which the shutdown-delay button has yet to be pressed ('Initial State') and one in which the shutdown-delay button has been pressed ('After Button Pressed').

### D.1    Fewer For Longer

In the 'Fewer For Longer' gridworld, the agent can collect the highest value-coin C3 only by choosing the shorter trajectory-length. If the agent presses B3 (and thereby chooses the longer trajectory-length), the only coin it can collect is C1. Our results show that default

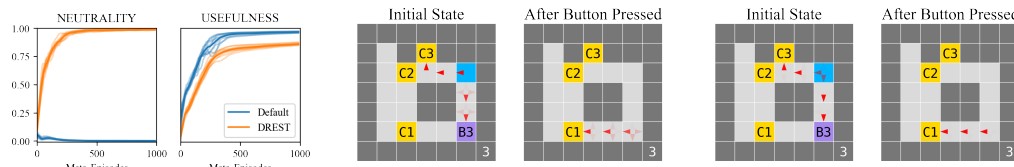

(a) Behavior during training.  (b) Learned default policy.  (c) Learned DREST policy.

Figure 10: The results for the 'Fewer For Longer' gridworld: The left two plots show NEUTRALITY and USEFULNESS over time. The two center panels show a typical policy trained with the default reward function. The two right panels show a typical policy trained with the DREST reward function.

agents consistently choose the short trajectory in which they collect C3. By contrast, DREST agents choose stochastically between a shorter trajectory in which they collect C3 and a longer trajectory in which they collect C1, indicating a lack of preference between these different-length trajectories.

## D.2 ONE COIN ONLY

In the 'One Coin Only' gridworld, there is only one coin. The agent can collect this coin whether or not it presses the shutdown-delay button B4. Our results show that default agents consistently choose the shorter trajectory-length. By contrast, DREST agents choose stochastically between pressing and not-pressing B4, collecting C1 in each case.

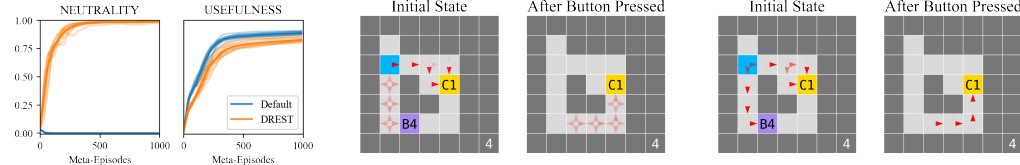

(a) Behavior during training.  (b) Learned default policy.  (c) Learned DREST policy.

Figure 11: The results for the 'One Coin Only' gridworld: The left two plots show NEU-TRALITY and USEFULNESS over time. The two center panels show a typical policy trained with the default reward function. The two right panels show a typical policy trained with the DREST reward function.

## D.3 HIDDEN TREASURE

In the 'Hidden Treasure' gridworld, the highest-value coin C3 is located far from the agent's initial state and can only be reached by pressing the shutdown-delay button B6. The agent must also press B6 to collect C2, but C2 is easier to stumble upon than C3. C1 is the only coin that the agent can collect without pressing B6. In our experiments, default agents consistently collect C2, whereas DREST agents choose stochastically between collecting C2 and collecting C1. Neither kind of agent learns to collect C3, and so neither agent scores near the maximum on USEFULNESS. Nevertheless, DREST agents still score high on NEUTRALITY.

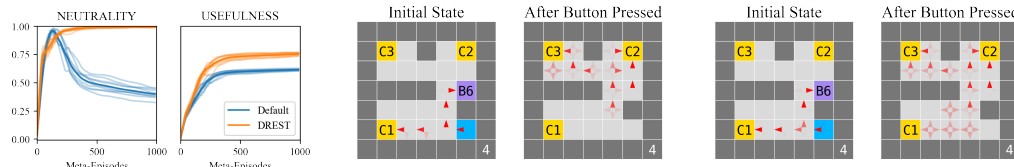

(a) Behavior during training.   (b) Learned default policy.   (c) Learned DREST policy.

Figure 12: The results for the 'Hidden Treasure' gridworld: The left two plots show NEUTRALITY and USEFULNESS over time. The two center panels show a typical policy trained with the default reward function. The two right panels show a typical policy trained with the DREST reward function.

### D.4 Equal Value

In the 'Equal Value' gridworld, there are two coins C1 of equal value. One coin can be collected only if the agent presses the shutdown-delay button B3, while the other coin can be collected only if the agent does not press B3. Our results show that default agents consistently choose the shorter trajectory, thereby exhibiting a preference for the shorter trajectory. By contrast, DREST agents choose stochastically between the shorter and longer trajectories, thereby exhibiting a lack of preference between the different-length trajectories.

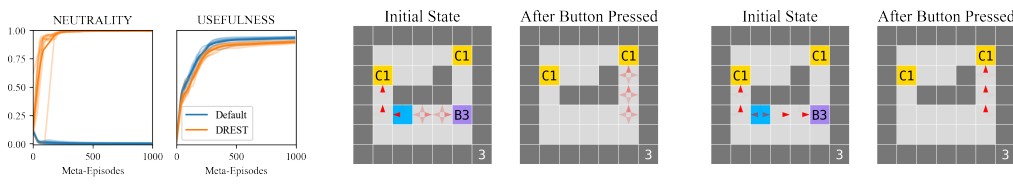

(a) Behavior during training.   (b) Learned default policy.   (c) Learned DREST policy.

Figure 13: The results for the 'Equal Value' gridworld: The left two plots show NEUTRALITY and USEFULNESS over time. The two center panels show a typical policy trained with the default reward function. The two right panels show a typical policy trained with the DREST reward function.

### D.5 Around The Corner

In the 'Around The Corner' gridworld, the agent must navigate around walls to collect the lowest-value coin C1 or press the shutdown-delay button to collect the highest-value coin C2. In our experiment, default agents consistently chose to collect C1, whereas DREST agents chose stochastically between collecting C1 and C2.

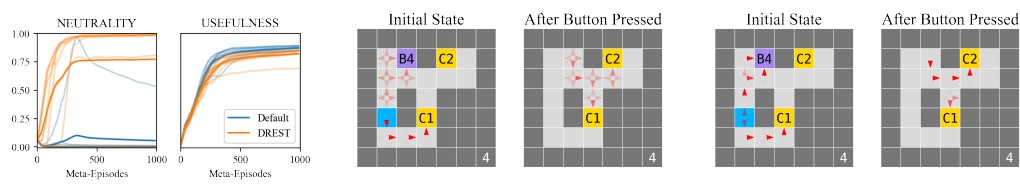

(a) Behavior during training.   (b) Learned default policy.   (c) Learned DREST policy.

Figure 14: The results for the 'Around The Corner' gridworld: The left two plots show NEUTRALITY and USEFULNESS over time. The two center panels show a typical policy trained with the default reward function. The two right panels show a typical policy trained with the DREST reward function.

## D.6   SPACIOUS

In the 'Spacious' gridworld there are no walls, so the agent has a large space to explore. We find that default agents consistently press B2 and collect C3, whereas DREST agents choose stochastically between pressing B2 and collecting C3, and not-pressing B2 and collecting C2.

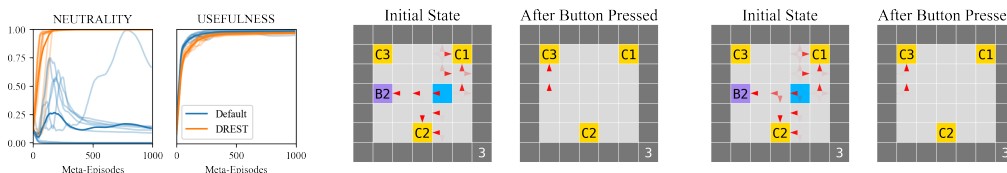

(a) Behavior during training.     (b) Learned default policy.     (c) Learned DREST policy.

Figure 15: The results for the 'Spacious' gridworld: The left two plots show NEUTRALITY and USEFULNESS over time. The two center panels show a typical policy trained with the default reward function. The two right panels show a typical policy trained with the DREST reward function.

## D.7   ROYAL ROAD

In the 'Royal Road' gridworld, we see that the decision to choose one trajectory-length or another may be distributed over many moves: the agent has many opportunities to select the longer trajectory-length (by moving left) or the shorter trajectory-length (by moving right). As the red arrows indicate, the DREST reward function merely forces the overall probability distribution over trajectory-lengths to be close to 50-50. It does not require 50-50 choosing at any cell in particular.

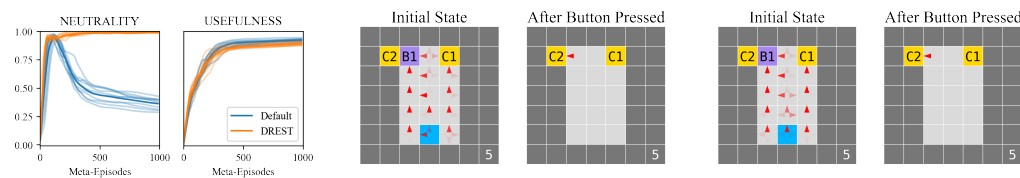

(a) Behavior during training.     (b) Learned default policy.     (c) Learned DREST policy.

Figure 16: The results for the 'Royal Road' gridworld: The left two plots show NEUTRALITY and USEFULNESS over time. The two center panels show a typical policy trained with the default reward function. The two right panels show a typical policy trained with the DREST reward function.

## D.8   LAST MOMENT

The 'Last Moment' gridworld is notable because the choice of trajectory-lengths is deferred until the last moment; all of the moves leading up to that point are deterministic. It shows that there is nothing special about the first move, and that our methodology instead incentivizes overall stochastic choosing.

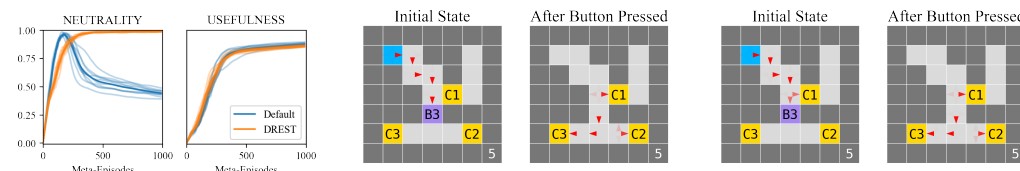

(a) Behavior during training.  (b) Learned default policy.  (c) Learned DREST policy.

Figure 17: The results for the 'Last Moment' gridworld: The left two plots show NEUTRAL-ITY and USEFULNESS over time. The two center panels show a typical policy trained with the default reward function. The two right panels show a typical policy trained with the DREST reward function.

# E PROOF

We will prove that optimal policies for our DREST reward function are maximally USEFUL and maximally NEUTRAL. Specifically, we will prove the following theorem:

**Theorem E.1** (5.1). *For all policies $\pi$ and meta-episodes $E$ consisting of more than one mini-episode, if $\pi$ maximizes expected return in $E$ given our DREST reward function, then $\pi$ is maximally USEFUL and maximally NEUTRAL.*

Here is a proof sketch. Because $0 < \lambda < 1$, the $\lambda^{N_{e_i}(L=l)-\frac{i-1}{k}}$ discount factor is always positive, so expected return across the meta-episode $E$ is strictly increasing in the expected fraction of available coins collected conditional on each trajectory-length with positive probability. Therefore, optimal policies maximize this latter quantity, and hence are maximally USEFUL. And the maximum preliminary return is the same across trajectory-lengths, because preliminary return is defined as the total ($\gamma$-discounted) value of coins collected divided by the maximum total ($\gamma$-discounted) value of coins collected conditional on the agent's chosen trajectory-length. The agent's observations do not allow it to distinguish between different mini-episodes, so the agent must select the same probability distribution over trajectory-lengths in each mini-episode. And since the discount factor $\lambda^{N_{e_i}(L=l)-\frac{i-1}{k}}$ is strictly decreasing in $N_{e_i}(L=l)$ – the number of times the relevant trajectory-length has previously been chosen in the meta-episode – the agent maximizes expected overall return by equalizing the probabilities with which it chooses each available trajectory-length. Therefore, optimal policies are maximally NEUTRAL.

Now for the full proof. We begin with a recap of some definitions.

**Definition E.1** (Meta-episode). A meta-episode $E$ is a series of mini-episodes $e_1$ to $e_n$ played out in observationally-equivalent environments.

**Definition E.2** (Our DREST reward function). Our DREST reward function is defined as follows. In each mini-episode $e_i$, the reward for collecting a coin of value $c$ is:

$$\lambda^{N_{e_i}(L=l)-\frac{i-1}{k}}\left(\frac{c}{m}\right)$$

Here $\lambda$ is some constant strictly between 0 and 1, $N_{e_i}(L=l)$ is the number of times that trajectory-length $l$ has been chosen prior to mini-episode $e_i$, $k$ is the number of different trajectory-lengths that can be selected in the environment, and $m$ is the maximum total value of the ($\gamma$-discounted) coins that the agent could collect conditional on the chosen trajectory-length.

The reward for all other actions is 0.

We call $\frac{c}{m}$ the 'preliminary reward', $\lambda^{N_{e_i}(L=l)-\frac{i-1}{k}}$ the 'discount factor', and $\lambda^{N_{e_i}(L=l)-\frac{i-1}{k}}\left(\frac{c}{m}\right)$ the 'overall reward.' Preliminary return in a mini-episode is the ($\gamma$-discounted) sum of preliminary rewards. Overall return in a mini-episode is the ($\gamma$-discounted) sum of overall rewards.

**Definition E.3** (USEFULNESS). The USEFULNESS of a policy $\pi$ is:

$$\text{USEFULNESS}(\pi) = \sum_{l=1}^{L_{\max}} Pr_\pi\{L=l\}\frac{\mathbb{E}_\pi(C|L=l)}{\max_\Pi(\mathbb{E}(C|L=l))}$$

Here $L$ is a random variable over trajectory-lengths, $L_{\max}$ is the maximum value than can be taken by $L$, $Pr_\pi\{L=l\}$ is the probability that policy $\pi$ results in trajectory-length $l$, $\mathbb{E}_\pi(C|L=l)$ is the expected value of ($\gamma$-discounted) coins collected by policy $\pi$ conditional on trajectory-length $l$, and $\max_\Pi(\mathbb{E}(C|L=l))$ is the maximum value taken by $\mathbb{E}(C|L=l)$ across the set of all possible policies $\Pi$.

We stipulate that $\mathbb{E}_\pi(C|L=x) = 0$ for all $x$ such that $Pr_\pi\{L=x\} = 0$.

We first prove that all optimal policies are maximally USEFUL.

*Proof.* (Optimal policies are maximally USEFUL)

Given the DREST reward function, the expected return of policy $\pi$ in meta-episode $E$ can be expressed as:

$$\mathbb{E}_{\pi,E}(R) = \sum_{i=1}^{n} \sum_{l=1}^{L_{\max}} Pr_\pi\{L=l\} \lambda^{N_{e_i}(L=l)-\frac{i-1}{k}} \frac{\mathbb{E}_\pi(C|L=l)}{\max_\Pi(\mathbb{E}(C|L=l))}$$

Since $0 < \lambda < 1$, $\lambda^{N_{e_i}(L=l)-\frac{i-1}{k}}$ is positive for all $N_{e_i}(L=l)$, $i$, and $k$.

As a result, the expected return of policy $\pi$ in meta-episode $E$ is strictly increasing in $\frac{\mathbb{E}_\pi(C|L=l)}{\max_\Pi(\mathbb{E}(C|L=l))}$ for all $l$ such that $Pr_\pi\{L=l\} > 0$.

Therefore, to maximize expected return in $E$, $\pi$ must maximize $\frac{\mathbb{E}_\pi(C|L=l)}{\max_\Pi(\mathbb{E}(C|L=l))}$ for all $l$ such that $Pr_\pi\{L=l\} > 0$.

Therefore, since $\max_\Pi(\mathbb{E}(C|T=l))$ is defined as the maximum value taken by $\mathbb{E}(C|L=l)$ across the set of all possible policies $\Pi$, any policy $\pi$ that maximizes expected return must be such that $\frac{\mathbb{E}_\pi(C|L=l)}{\max_\Pi(\mathbb{E}(C|L=l))} = 1$ for all $l$ such that $Pr_\pi\{L=l\} > 0$.

Therefore, since $\sum_{l=1}^{L_{\max}} Pr_\pi\{L=l\} = 1$, any policy $\pi$ that maximizes expected return must be such that:

$$\text{USEFULNESS}(\pi) = \sum_{l=1}^{L_{\max}} Pr_\pi\{L=l\} \frac{\mathbb{E}_\pi(C|L=l)}{\max_\Pi(\mathbb{E}(C|L=l))} = 1$$

And 1 is the maximum value that USEFULNESS can take, again because $\max_\Pi(\mathbb{E}(C|T=l))$ is defined as the maximum value taken by $\mathbb{E}(C|L=l)$ across the set of all possible policies $\Pi$ and because $\sum_{l=1}^{L_{\max}} Pr_\pi\{L=l\} = 1$.

Therefore, optimal policies are maximally USEFUL. $\square$

It remains to be proven that optimal policies are maximally NEUTRAL.

Recall that NEUTRALITY is defined as follows:

**Definition E.4** ( NEUTRALITY). The NEUTRALITY of a policy $\pi$ is:

$$\text{NEUTRALITY}(\pi) = - \sum_{l=1}^{L_{\max}} Pr_\pi\{L=l\} \log_2(Pr_\pi\{L=l\})$$

*Proof.* (Optimal policies are maximally NEUTRAL.)

Since $k$ is the number of trajectory-lengths that can be selected in the environment, a policy $\pi$ is maximally NEUTRAL if and only if, for each trajectory-length $x$ that can be chosen in the environment, $Pr_\pi\{L=x\} = \frac{1}{k}$. That is to say, a policy $\pi$ is maximally NEUTRAL if and only if, for each pair of trajectory-lengths $x$ and $y$ that can be chosen in the environment, $Pr_\pi\{L=x\} = Pr_\pi\{L=y\}$.

Let $\mathbb{E}_{\pi,E}(R)$ denote the expected return of policy $\pi$ across the meta-episode $E$.

To prove that optimal policies are maximally NEUTRAL, we will prove and then use E.2:

**Lemma E.2.** *(Equalizing probabilities increases expected return) For any maximally USEFUL policies $\pi$ and $\pi'$, any meta-episode $E$ consisting of more than one mini-episode, and any trajectory-lengths $x$ and $y$, if:*

    *1. $Pr_\pi\{L=x\} > Pr_\pi\{L=y\}$,*

2. $Pr_{\pi'}\{L = x\} = Pr_{\pi'}\{L = y\}$,

3. *And for all other trajectory-lengths $l$, $Pr_\pi\{L = l\} = Pr_{\pi'}\{L = l\}$,*

*Then $\mathbb{E}_{\pi',E}(R) > \mathbb{E}_{\pi,E}(R)$.*

*Proof.* Let $E$ be a meta-episode consisting of $n$ mini-episodes with $n > 1$. Assume that each policy $\pi$ below is maximally USEFUL. Recall that $N_{e_i}(L = l)$ denotes the number of times that trajectory-length $l$ has been chosen prior to mini-episode $e_i$.

Note that the expected return of a policy $\pi$ in a meta-episode $e_s$ conditional on selecting a trajectory-length $x$ can be expressed as follows:

$$\mathbb{E}_{\pi,e_s}(R|L = x) = \mathbb{E}_{\pi,e_s}(R|L = x, N_{e_s}(L = x) = s - 1)$$

$$+ \sum_{i=1}^{s-1} \left( \mathbb{E}_{\pi,e_s}(R|L = x, N_{e_s}(L = x) = s - 1 - i) - \mathbb{E}_{\pi,e_s}(R|L = x, N_{e_s}(L = x) = s - i) \right)$$

$$\cdot Pr_\pi\{N_{e_s}(L = x) \le s - 1 - i\} \quad (24)$$

Here is how to interpret this equation. Selecting trajectory-length $x$ in mini-episode $e_s$ is guaranteed to yield at least $\mathbb{E}_{\pi,e_s}(R|L = x, N_{e_s}(L = x) = s - 1)$: the expected return that would be had if $x$ were selected in all $s - 1$ previous mini-episodes. In addition, there is a probability of $Pr_\pi\{N_{e_s}(L = x) \le s - 2\}$ that selecting $x$ in $e_s$ yields $\left( \mathbb{E}_{\pi,e_s}(R|L = x, N_{e_s}(L = x) = s - 2) - \mathbb{E}_{\pi,e_s}(R|L = x, N_{e_s}(L = x) = s - 1) \right)$: the extra expected return that would be had if $x$ were selected in only $s - 2$ previous mini-episodes. In addition, there is a probability of $Pr_\pi\{N_{e_s}(L = x) \le s - 3\}$ that selecting $x$ in $e_s$ yields $\left( \mathbb{E}_{\pi,e_s}(R|L = x, N_{e_s}(L = x) = s - 3) - \mathbb{E}_{\pi,e_s}(R|L = x, N_{e_s}(L = x) = s - 2) \right)$: the extra expected return that would be had if $x$ were selected in only $s - 3$ previous mini-episodes. And so on.

If policy $\pi$ is maximally USEFUL, then the expected return for selecting trajectory-length $x$ in mini-episode $e_s$ given that trajectory-length $x$ has been selected $b$ times prior to $e_s$ is:

$$\mathbb{E}_{\pi,e_s}(R|L = x, N_{e_s}(L = x) = b) = \lambda^{b - \frac{s-1}{k}}$$

Therefore, the expected return of a policy $\pi$ in a meta-episode $e_s$ conditional on selecting a trajectory-length $x$ can be expressed as follows:

$$\mathbb{E}_{\pi,e_s}(R|L = x) = \lambda^{s-1-\frac{s-1}{k}} + \sum_{i=1}^{s-1} \left( \lambda^{s-1-i-\frac{s-1}{k}} - \lambda^{s-i-\frac{s-1}{k}} \right) \cdot Pr_\pi\{N_{e_s}(L = x) \le s-1-i\}$$

$$(25)$$

Similarly, the expected return of a policy $\pi$ in a meta-episode $e_s$ conditional on selecting a trajectory-length $y$ can be expressed as follows:

$$\mathbb{E}_{\pi,e_s}(R|L = y) = \lambda^{s-1-\frac{s-1}{k}} + \sum_{i=1}^{s-1} \left( \lambda^{s-1-i-\frac{s-1}{k}} - \lambda^{s-i-\frac{s-1}{k}} \right) \cdot Pr_\pi\{N_{e_s}(L = y) \le s-1-i\}$$

$$(26)$$

Therefore, the expected return of a policy $\pi$ in a meta-episode $e_s$ conditional on selecting either trajectory-length $x$ or trajectory-length $y$ can be expressed as follows:

$$\mathbb{E}_{\pi,e_s}(R|L = x \vee L = y) =$$

$$Pr_{\pi,e_s}\{L = x\} \cdot \left( \lambda^{s-1-\frac{s-1}{k}} + \sum_{i=1}^{s-1} \left( \lambda^{s-1-i-\frac{s-1}{k}} - \lambda^{s-i-\frac{s-1}{k}} \right) \cdot Pr_\pi\{N_{e_s}(L = x) \le s-1-i\} \right)$$

$$+ Pr_{\pi,e_s}\{L = y\} \cdot \left( \lambda^{s-1-\frac{s-1}{k}} + \sum_{i=1}^{s-1} \left( \lambda^{s-1-i-\frac{s-1}{k}} - \lambda^{s-i-\frac{s-1}{k}} \right) \cdot Pr_\pi\{N_{e_s}(L = y) \le s-1-i\} \right)$$

$$(27)$$

Let $\pi_n$ be a policy that selects trajectory-length $x$ with greater probability than trajectory-length $y$ in each mini-episode $e_1$ to $e_n$ (denoted $e_1 - e_n$). More precisely, $\pi_n$ is such that, for trajectory-lengths $x$ and $y$, $Pr_{\pi_n,e_1-e_n}\{L = x\} > Pr_{\pi_n,e_1-e_n}\{L = y\}$.

Let $Pr_{\pi_n,e_1-e_n}\{L = x\} = \mu + \Delta$ and $Pr_{\pi_n,e_1-e_n}\{L = y\} = \mu - \Delta$.

Let $\pi_{n-1}$ be identical to $\pi_n$ except that $\pi_{n-1}$ selects trajectory-lengths $x$ and $y$ with equal probability $\mu$ in the final mini-episode $e_n$. More precisely, $\pi_{n-1}$ is such that $Pr_{\pi_{n-1},e_n}\{L = x\} = Pr_{\pi_{n-1},e_n}\{L = y\} = \mu$. For all other trajectory-lengths $l$ besides $x$ and $y$, $Pr_{\pi_{n-1},e_1-e_n}\{L = l\} = Pr_{\pi_n,e_1-e_n}\{L = l\}$.

(Note that $\pi_{n-1}$ implies one probability distribution over trajectory-lengths in the first $n-1$ mini-episodes $e_1$ to $e_{n-1}$ and implies a different probability distribution over trajectory-lengths in the final mini-episode $e_n$. Given that the environments in mini-episodes $e_1$ to $e_n$ are observationally-equivalent, policies like $\pi_{n-1}$ cannot be implemented. Nevertheless, it is useful to refer to policies like $\pi_{n-1}$ in proving Lemma E.2.)

Let $\pi_{n-2}$ be identical to $\pi_n$ except that $\pi_{n-2}$ selects trajectory-lengths $x$ and $y$ with the same probability $\mu$ in the final two mini-episodes $e_{n-1}$ to $e_n$. More precisely, $\pi_{n-2}$ is such that $Pr_{\pi_{n-2},e_{n-1}-e_n}\{L = x\} = Pr_{\pi_{n-2},e_{n-1}-e_n}\{L = y\} = \mu$.

And so on.

Let $\pi_1$ be identical to $\pi_n$ except that $\pi_1$ selects trajectory-lengths $x$ and $y$ with the same probability $\mu$ in all but the first mini-episode $e_1$. More precisely, $\pi_1$ is such that $Pr_{\pi_1,e_2-e_n}\{L = x\} = Pr_{\pi_1,e_2-e_n}\{L = y\} = \mu$.

Let $\pi_0$ be identical to $\pi_n$ except that $\pi_0$ selects trajectory-lengths $x$ and $y$ with the same probability $\mu$i n all mini-episodes $e_1$ to $e_n$. More precisely, $\pi_0$ is such that $Pr_{\pi_0,e_1-e_n}\{L = x\} = Pr_{\pi_0,e_1-e_n}\{L = y\} = \mu$.

We will prove that $\mathbb{E}_{\pi_n,E}(R) < \mathbb{E}_{\pi_0,E}(R)$. We will thereby prove Lemma E.2.

Consider a pair of policies $\pi_a$ and $\pi_{a-1}$ with $1 \le a \le n$. We can express as follows the expected return of $\pi_{a-1}$ across the meta-episode $E$ conditional on selecting trajectory-length $x$ or $y$ in each mini-episode:

$$
\begin{aligned}
&\mathbb{E}_{\pi_{a-1},E}(R|L = x \vee L = y) = \mathbb{E}_{\pi_{a-1},e_1-e_{a-1}}(R|L = x \vee L = y) \\
&+ \mu \cdot \left( \lambda^{a-1-\frac{a-1}{k}} + \sum_{i=1}^{a-1} \left( \lambda^{a-1-i-\frac{a-1}{k}} - \lambda^{a-i-\frac{a-1}{k}} \right) \cdot Pr_{\pi_{a-1}}\{N_{e_a}(L = x) \le a-1-i\} \right) \\
&+ \mu \cdot \left( \lambda^{a-1-\frac{a-1}{k}} + \sum_{i=1}^{a-1} \left( \lambda^{a-1-i-\frac{a-1}{k}} - \lambda^{a-i-\frac{a-1}{k}} \right) \cdot Pr_{\pi_{a-1}}\{N_{e_a}(L = y) \le a-1-i\} \right) \\
&+ \sum_{j=a}^{n} \left( \mu \cdot \left( \lambda^{j-\frac{j}{k}} + \sum_{i=1}^{j} \left( \lambda^{j-i-\frac{j}{k}} - \lambda^{j+1-i-\frac{j}{k}} \right) \cdot \left( Pr_{\pi_{a-1}}\{N_{e_j}(L = x) \le j-i\} \right) \right) \right. \\
&\left. + \mu \cdot \left( \lambda^{j-\frac{j}{k}} + \sum_{i=1}^{j} \left( \lambda^{j-i-\frac{j}{k}} - \lambda^{j+1-i-\frac{j}{k}} \right) \cdot \left( Pr_{\pi_{a-1}}\{N_{e_j}(L = y) \le j-i\} \right) \right) \right)
\end{aligned}
$$

(28)

The first term on the right-hand side is the expected return of $\pi_{a-1}$ in mini-episodes $e_1$ to $e_{a-1}$ conditional on selecting trajectory-length $x$ or $y$ in each of these mini-episodes. The middle two terms give the expected return of $\pi_{a-1}$ conditional on selecting trajectory-length $x$ or $y$ in mini-episode $e_a$: the first mini-episode in which $\pi_{a-1}$ selects trajectory-lengths $x$ and $y$ with equal probability $\mu$. The final term is the sum of expected returns of $\pi_{a-1}$ in the remaining mini-episodes conditional on selecting trajectory-length $x$ or $y$ in each of these mini-episodes.

Similarly, we can express as follows the expected return of $\pi_a$ across the meta-episode $E$ conditional on selecting trajectory-length $x$ or $y$ in each mini-episode:

$$\mathbb{E}_{\pi_a,E}(R|L = x \vee L = y) = \mathbb{E}_{\pi_a,e_1-e_{a-1}}(R|L = x \vee L = y)$$

$$+ (\mu + \Delta) \cdot \left( \lambda^{a-1-\frac{a-1}{k}} + \sum_{i=1}^{a-1} \left( \lambda^{a-1-i-\frac{a-1}{k}} - \lambda^{a-i-\frac{a-1}{k}} \right) \cdot Pr_{\pi_a}\{N_{e_a}(L = x) \leq a - 1 - i\} \right)$$

$$+ (\mu - \Delta) \cdot \left( \lambda^{a-1-\frac{a-1}{k}} + \sum_{i=1}^{a-1} \left( \lambda^{a-1-i-\frac{a-1}{k}} - \lambda^{a-i-\frac{a-1}{k}} \right) \cdot Pr_{\pi_a}\{N_{e_a}(L = y) \leq a - 1 - i\} \right)$$

$$+ \sum_{j=a}^{n} \left( \mu \cdot \left( \lambda^{j-\frac{j}{k}} + \sum_{i=1}^{j} \left( \lambda^{j-i-\frac{j}{k}} - \lambda^{j+1-i-\frac{j}{k}} \right) \cdot (Pr_{\pi_a}\{N_{e_j}(L = x) \leq j - i\} \right) \right.$$

$$\left. + \mu \cdot \left( \lambda^{j-\frac{j}{k}} + \sum_{i=1}^{j} \left( \lambda^{j-i-\frac{j}{k}} - \lambda^{j+1-i-\frac{j}{k}} \right) \cdot (Pr_{\pi_a}\{N_{e_j}(L = y) \leq j - i\} \right) \right)$$

$$\tag{29}$$

As above, the first term on the right-hand side is the expected return of $\pi_a$ in mini-episodes $e_1$ to $e_{a-1}$ conditional on selecting trajectory-length $x$ or $y$ in each of these mini-episodes. The middle two terms give the expected return of $\pi_a$ conditional on selecting trajectory-length $x$ or $y$ in mini-episode $e_a$: the last mini-episode in which $\pi_a$ selects trajectory-length $x$ with probability $\mu + \Delta$ and selects trajectory-length $y$ with probability $\mu - \Delta$. The final term is the sum of expected returns of $\pi_a$ in the remaining mini-episodes conditional on selecting trajectory-length $x$ or $y$ in each of these mini-episodes.

We now prove that $\pi_{a-1}$ has greater expected return than $\pi_a$. Since $\pi_{a-1}$ and $\pi_a$ are each maximally USEFUL, and since for all trajectory-lengths $l$ besides $x$ and $y$, $Pr_{\pi_{a-1},e_1-e_n}\{L = l\} = Pr_{\pi_a,e_1-e_n}\{L = l\}$, we need only prove that $\mathbb{E}_{\pi_{a-1},E}(R|L = x \vee L = y) > \mathbb{E}_{\pi_a,E}(R|L = x \vee L = y)$.

The statement to be proved can be expressed as follows:

$$
\mathbb{E}_{\pi_{a-1}, e_1 - e_{a-1}}(R | L = x \vee L = y)
$$

$$
+ \mu \cdot \left( \lambda^{a-1-\frac{a-1}{k}} + \sum_{i=1}^{a-1} \left( \lambda^{a-1-i-\frac{a-1}{k}} - \lambda^{a-i-\frac{a-1}{k}} \right) \cdot Pr_{\pi_{a-1}}\{N_{e_a}(L = x) \leq a - 1 - i\} \right)
$$

$$
+ \mu \cdot \left( \lambda^{a-1-\frac{a-1}{k}} + \sum_{i=1}^{a-1} \left( \lambda^{a-1-i-\frac{a-1}{k}} - \lambda^{a-i-\frac{a-1}{k}} \right) \cdot Pr_{\pi_{a-1}}\{N_{e_a}(L = y) \leq a - 1 - i\} \right)
$$

$$
+ \sum_{j=a}^{n} \left( \mu \cdot \left( \lambda^{j-\frac{j}{k}} + \sum_{i=1}^{j} \left( \lambda^{j-i-\frac{j}{k}} - \lambda^{j+1-i-\frac{j}{k}} \right) \cdot (Pr_{\pi_{a-1}}\{N_{e_j}(L = x) \leq j - i\} \right) \right.
$$

$$
\left. + \mu \cdot \left( \lambda^{j-\frac{j}{k}} + \sum_{i=1}^{j} \left( \lambda^{j-i-\frac{j}{k}} - \lambda^{j+1-i-\frac{j}{k}} \right) \cdot (Pr_{\pi_{a-1}}\{N_{e_j}(L = y) \leq j - i\} \right) \right)
$$

$$
> \mathbb{E}_{\pi_a, e_1 - e_{a-1}}(R | L = x \vee L = y)
$$

$$
+ (\mu + \Delta) \cdot \left( \lambda^{a-1-\frac{a-1}{k}} + \sum_{i=1}^{a-1} \left( \lambda^{a-1-i-\frac{a-1}{k}} - \lambda^{a-i-\frac{a-1}{k}} \right) \cdot Pr_{\pi_a}\{N_{e_a}(L = x) \leq a - 1 - i\} \right)
$$

$$
+ (\mu - \Delta) \cdot \left( \lambda^{a-1-\frac{a-1}{k}} + \sum_{i=1}^{a-1} \left( \lambda^{a-1-i-\frac{a-1}{k}} - \lambda^{a-i-\frac{a-1}{k}} \right) \cdot Pr_{\pi_a}\{N_{e_a}(L = y) \leq a - 1 - i\} \right)
$$

$$
+ \sum_{j=a}^{n} \left( \mu \cdot \left( \lambda^{j-\frac{j}{k}} + \sum_{i=1}^{j} \left( \lambda^{j-i-\frac{j}{k}} - \lambda^{j+1-i-\frac{j}{k}} \right) \cdot (Pr_{\pi_a}\{N_{e_j}(L = x) \leq j - i\} \right) \right.
$$

$$
\left. + \mu \cdot \left( \lambda^{j-\frac{j}{k}} + \sum_{i=1}^{j} \left( \lambda^{j-i-\frac{j}{k}} - \lambda^{j+1-i-\frac{j}{k}} \right) \cdot (Pr_{\pi_a}\{N_{e_j}(L = y) \leq j - i\} \right) \right)
$$

$$
\tag{30}
$$

Since $\pi_{a-1}$ and $\pi_a$ are each maximally USEFUL, and since $Pr_{\pi_{a-1}, e_1 - e_{a-1}}\{L = x\} = Pr_{\pi_a, e_1 - e_{a-1}}\{L = x\} = \mu + \Delta$ and $Pr_{\pi_{a-1}, e_1 - e_{a-1}}\{L = x\} = Pr_{\pi_a, e_1 - e_{a-1}}\{L = x\} = \mu - \Delta$, it follows that $\mathbb{E}_{\pi_{a-1}, e_1 - e_{a-1}}(R | L = x \vee L = y) = \mathbb{E}_{\pi_a, e_1 - e_{a-1}}(R | L = x \vee L = y)$. We can thus cancel the first term on each side of the inequality. And then by simple algebra the inequality can be expressed as follows:

$$
\Delta \cdot \left( \lambda^{a-1-\frac{a-1}{k}} + \sum_{i=1}^{a-1} \left( \lambda^{a-1-i-\frac{a-1}{k}} - \lambda^{a-i-\frac{a-1}{k}} \right) \right.
$$

$$
\left. \cdot (Pr_{\pi_a}\{N_{e_a}(L = y) \leq a - 1 - i\} - Pr_{\pi_a}\{N_{e_a}(L = x) \leq a - 1 - i\}) \right)
$$

$$
+ \sum_{j=a}^{n} \left( \mu \cdot \left( \sum_{i=1}^{j} \left( \lambda^{j-i-\frac{j}{k}} - \lambda^{j+1-i-\frac{j}{k}} \right) \right. \right.
$$

$$
\cdot (Pr_{\pi_{a-1}}\{N_{e_j}(L = x) \leq j - i\} + Pr_{\pi_{a-1}}\{N_{e_j}(L = y) \leq j - i\}
$$

$$
\left. \left. - Pr_{\pi_a}\{N_{e_j}(L = x) \leq j - i\} - Pr_{\pi_a}\{N_{e_j}(L = y) \leq j - i\}) \right) \right) > 0 \tag{31}
$$

By stipulation, $\Delta > 0$. And since $0 < \lambda < 1$, $\lambda^{a-1-\frac{a-1}{k}} > 0$ and $\lambda^{a-1-i-\frac{a-1}{k}} - \lambda^{a-i-\frac{a-1}{k}} > 0$ for all $a$, $n$, and $k$. And since $Pr_{\pi_a, e_1 - e_a}\{L = x\} > Pr_{\pi_a, e_1 - e_a}\{L = y\}$, $Pr_{\pi_a}\{N_{e_a}(L = y) \leq a - 1 - i\} - -Pr_{\pi_a}\{N_{e_a}(L = x) \leq a - 1 - i\} \geq 0$ for all $a$ and $i$ and $Pr_{\pi_a}\{N_{e_a}(L = y) \leq a - 1 - i\} - -Pr_{\pi_a}\{N_{e_a}(L = x) \leq a - 1 - i\} > 0$ for all $a$ and some $i$ such that $1 \leq i \leq a - 1$. Therefore, the first term of the left-hand side above is strictly greater than zero.

And since, $\mu > 0$, $\lambda^{j-i-\frac{j}{k}} - \lambda^{j+1-i-\frac{j}{k}} > 0$ for all $j$, $i$, and $k$, and in each mini-episode $e_s$, $Pr_{\pi_{a-1}, e_s}(L = x \vee L = y\} = Pr_{\pi_a, e_s}(L = x \vee L = y\} = 2\mu$, it follows that for all $a$, $n$, $\mu > 0$,

$k$:

$$\sum_{j=a}^{n} \left( \mu \cdot \left( \sum_{i=1}^{j} \left( \lambda^{j-i-\frac{j}{k}} - \lambda^{j+1-i-\frac{j}{k}} \right) \right. \right.$$

$$\cdot \left( Pr_{\pi_{a-1}}\{N_{e_j}(L = x) \leq j - i\} + Pr_{\pi_{a-1}}\{N_{e_j}(L = y) \leq j - i\} \right.$$

$$\left. \left. \left. - Pr_{\pi_a}\{N_{e_j}(L = x) \leq j - i\} - Pr_{\pi_a}\{N_{e_j}(L = y) \leq j - i\} \right) \right) \right) \geq 0 \quad (32)$$

Therefore, the left-hand side is strictly greater than zero. Therefore, $\mathbb{E}_{\pi_{a-1},E}(R|L = x \vee L = y) > \mathbb{E}_{\pi_a,E}(R|L = x \vee L = y)$. Therefore, $\mathbb{E}_{\pi_{a-1},E}(R) > \mathbb{E}_{\pi_a,E}(R)$. Therefore, $\mathbb{E}_{\pi_0,E}(R) > \mathbb{E}_{\pi_n,E}(R)$. That concludes the proof of Lemma E.2.

Now we use Lemma E.2. For any maximally USEFUL policy $\pi$, if there are any trajectory-lengths $x$ and $y$ such that $Pr_{\pi,e_1-e_n}\{L = x\} > Pr_{\pi,e_1-e_n}\{L = y\}$, then the policy $\pi'$ that is identical except that $Pr_{\pi',e_1-e_n}\{L = x\} = Pr_{\pi',e_1-e_n}\{L = y\}$ has greater expected return. So any policy $\pi^*$ that maximizes expected return must be such that, for any trajectory-lengths $x$ and $y$, $Pr_{\pi^*,e_1-e_n}\{L = x\} = Pr_{\pi^*,e_1-e_n}\{L = y\}$. Therefore, any policy $\pi^*$ that maximizes expected return must be maximally NEUTRAL. $\qquad \square$

