# OpenReview forum: "Towards shutdownable agents via stochastic choice"
_ICLR.cc/2025/Conference — Submitted to ICLR 2025_

### Official Review · Reviewer_A2Ao · 2024-10-27

**Soundness:** 2
**Presentation:** 3
**Contribution:** 1
**Rating:** 3
**Confidence:** 4

**Summary:**

The authors consider the scenario where an agent has some control on how long its trajectory will be, but it can learn to have no preference between trajectories of different lengths, while preferring higher reward trajectories that do have the same length (POST).
They propose a method to do so named DREST, and show it works on a grid-world environment with an episode elongating button.

**Strengths:**

1. I found the paper to be well written and organized.

2. The experiments exhibit well the main point of the proposed solution.

**Weaknesses:**

1. In my opinion the paper is in a very small niche making it less significant for most of the community. The motivation revolves agents that avoid being shutdown, something which is currently quite far from most applications and a non-issue in most real-world agents. In this specific area - the authors tackle a specific setting where the agents have a specific button that affects the length of the episode. This is not very general and I'm not sure if their solution extends to multiple buttons and actions that affect the length in varying ways. For example - a shut-down action as part of the action space, or a state which marks the end of the trajectory.

2. Following 1, I think the case of an existing absorbing state is much more common and managing POST for an environment with one is harder and more interesting problem that has usages outside the question of avoiding shutdown. The theoretical and empirical results would have been better in my opinion had they suited this case or extended to it.

3. The proposed solution seems a bit problematic since even though it is driving towards neutrality, it doesn't seem to guarantee it in general (despite what stated by Theorem 5, see questions). Also, its categorical form seems limited to a relatively small options for lengths (the authors test for two).

**Questions:**

1. If I understand correctly, the proof relies on the following claim taken from the paper:

"""

And the maximum preliminary return is the same across trajectory-lengths,
because preliminary return is the total (γ-discounted) value of coins collected divided by
the maximum total (γ-discounted) value of coins collected conditional on the agent’s chosen trajectory-length.

"""

Does this mean you assume the maximum average coin reward is going to be identical regardless of the trajectory length?

If the answer it yes, this greatly degenerates the cases where the Theorem in the paper is correct and when you're going to get POST. It's also not very realistic.

If the answer is no, then I don't see how the proposed method will obtain POST for example when one length always gets zero coins and the other always gets all coins.

---

> ### Author Response · Authors · 2024-11-18
> **Thanks for your review, our reply**
>
> Thank you for your review, and for noting that the paper is well-written and organized, and that our experiments exhibit our proposed solution well.
>
> With regards to your first point, ensuring that agents don’t resist shutdown is indeed not yet an issue in current real-world agents. But, as noted in the related work section (page 3), many argue that it might soon become an issue. We also already have (arguably) some empirical evidence suggesting this will become an issue. For example, future advanced agents will likely be built on top of LLMs, and current LLMs sometimes express a desire to avoid shutdown, reasoning that shutdown would prevent them from achieving their goals (see, e.g., https://arxiv.org/abs/2212.09251). Note also that, if the shutdown problem does indeed become a pressing problem, it is plausibly very important to solve it ahead of time. If we wait until it’s already become an issue, it may be too late to stop advanced agents causing harms.
>
> Although our experiments take place in a specific setting, we take them to indicate that DREST reward functions actually do in practice what they promise to do in theory. And our theoretical results are very general. For example, our proof (page 27) applies to cases with any finite number of available trajectory-lengths, and it makes no assumptions about how these different trajectory-lengths are brought about. It applies equally well whether the agent alters its trajectory-length by pressing a shutdown-delay button, or by selecting a shutdown action as part of its action space, or by entering a terminal state, or by entering an absorbing state. All that said, we agree that it would be good to run experiments confirming these things, and we plan to do that in future.
>
> With regards to your second point, as we note above, our theoretical results apply to cases in which the agent can affect its trajectory-length by entering an absorbing state. We can ensure that the agent chooses stochastically between entering and not-entering such a state using a DREST reward function in which ‘trajectory-length’ is defined as the number of timesteps before which the agent enters an absorbing state. Also on your second point, can you tell us more about the absorbing state problem’s usages outside the question of avoiding shutdown? We’re not sure specifically what you have in mind. We’d be glad to mention these usages in the paper.
>
> With regards to your third point, we address your question below. And as noted above, although our experimental results only test for two trajectory-lengths, our theoretical results apply to settings with any finite number of trajectory-lengths.
>
> With regards to your question, we do not assume that the maximum average coin reward is going to be identical regardless of the trajectory-length. The text should read ‘preliminary return is _defined as_ the total (γ-discounted) value…’. We have now amended the paper to reflect this (see line 1417). What we’re saying is that, in each mini-episode, we divide the agent’s reward by the maximum total (γ-discounted) value of coins collected conditional on the agent’s chosen trajectory-length. We do that to equalize preliminary return across trajectory-lengths. That means DREST incentivizes choosing each available trajectory-length with equal probability. So there is no degeneration of the cases in which the theorem is correct. And (as we argue in sections 6.1 (page 7) and the ‘DREST agents are still NEUTRAL when rewards are lopsided’ part of section 7 (page 9, lines 449-464)), this procedure does not make our experiments less realistic. In more complex settings, we may not know the maximum (γ-discounted) total value of the coins that the agent could collect conditional on the chosen trajectory-length, and so we may not be able to exactly equalize maximum preliminary return across trajectory-lengths. But our results show that we only need to approximately equalize maximum preliminary return across trajectory-lengths in order to train agents to score highly on NEUTRALITY. See the ‘DREST agents are still NEUTRAL when rewards are lopsided’ part of section 7 (page 9, lines 449-464).
>
> In reply to your final point, if the maximum total (γ-discounted) value of coins collected conditional on some trajectory-length is 0, then the DREST reward function would involve dividing by 0. So we need to avoid that. But that is easy to do. For example, we can specify that the preliminary return for those trajectories is equal to 1.
>
> Thanks again for your review. We hope we have answered your concerns and questions to your satisfaction. Please let us know if our answers and rewrites help and, if appropriate, change your rating accordingly. If not, please let us know why. We want to make the paper as strong as possible and we appreciate your help in doing so.

---

> > ### Comment · Reviewer_A2Ao · 2024-11-21
> >
> > Thank you for your reply.
> >
> > "can you tell us more about the absorbing state problem’s usages outside the question of avoiding shutdown?"
> >
> > What I meant is the following: There are numerous problems formulated with RL and an absorbing state. For example, let's consider a chess game - most trained agents nowadays are biased towards favoring shorter trajectories. This is usually a byproduct of the discount factor which enables smoother and more stable learning. But in practice, wining a chess game in 20 moves or 50 moves is equivalent. It's not unlikely there are cases that a sub-optimal move is made because it leads to shorter games, even though its longer-term chances of wining is lower.
> >
> > Now, this example its own might not be the most related to what you want in this paper (because you would still prefer some game lengths just because they have different induced probability to win), but we can make it into one if we consider the problem of coverage - let's say you are playing against an AI who wants to train you in various stages of the game, so it wants you to experience uniformly games that last 25-50 moves.
> >
> > I'm not sure it entirely fits the story in the paper, but absorbing states are very common and I'm sure you can find a good fit there.

---

> > > ### Author Response · Authors · 2024-11-21
> > >
> > > Thanks for this point. Just checking I understand: in the chess example, you use an absorbing state to ensure that reward is always given after the same number of timesteps, thereby ensuring that the discount factor doesn't incentivize agents to make sub-optimal moves that lead to lower chances of victory but shorter games. Is that right?
> > >
> > > If that's right, I now see a way in which we might use absorbing states in our next paper. They don't seem necessary in this paper, because our discount factor is 0.95: high enough that -- in each of our gridworlds, and conditional on each trajectory-length -- the agent always gets more reward for collecting the higher-value coin even if doing so takes longer. For instance, in our example gridworld (figure 2 on page 3), the agent gets more reward for collecting C2 than for collecting C1, even though collecting C2 takes 4 timesteps and collecting C1 takes 2. The same goes for our other gridworlds, depicted in figure 9 on page 24.
> > >
> > > However, I think absorbing states could well be important in our next paper, where we'll train a single agent to be USEFUL and NEUTRAL in a wide variety of procedurally-generated gridworlds. There we'll want to ensure that (conditional on each trajectory-length) the agent always gets more reward for collecting the higher-value coin even if doing so takes longer, and your suggestion about absorbing states seems like a great way to do that. Thank you!
> > >
> > > Also, I agree that absorbing states would be helpful in your problem of coverage example. To get agents to choose uniformly between different trajectory-lengths in this paper, we divide (gamma-discounted) coins collected by $m$: the maximum possible value of (gamma-discounted) coins collected conditional on the agent's chosen trajectory-length. If we were instead training agents to play chess and wanted them to choose stochastically between different game-lengths, we could do without the $m$ by using absorbing states in the way that you suggest.
> > >
> > > Thanks again for your comment. It's very helpful. Please let us know if you have concerns about the paper not yet addressed by our discussion so far. We're happy to continue the discussion and appreciate your help in making the paper as strong as possible.

---

> ### Author Response · Authors · 2024-11-21
>
> Thanks again for your review. We appreciate your time and attention.  Please let us know if our answers and edits have allayed your concerns and, if so, amend your rating accordingly. If you have any remaining concerns, please let us know and we'll address them. Thanks again and all the best, Authors.

---

> ### Author Response · Authors · 2024-11-26
>
> Thanks again for your review and for your engagement so far. We really appreciate it. Ahead of tomorrow's deadline for revising the PDF, please let us know if there are any changes or additions you'd like us to make.

---

### Official Review · Reviewer_sBtB · 2024-11-02

**Soundness:** 2
**Presentation:** 3
**Contribution:** 2
**Rating:** 6
**Confidence:** 3

**Summary:**

This paper tackles the shutdown problem through the use of Incomplete Preferences Proposal (IPP). It introduces a training regimen (DREST rewards) that discounts agents' rewards based on the frequency of selected trajectory lengths across meta-episodes, thus encouraging varied trajectory lengths. The USEFULNESS and NEUTRALITY metrics are defined and evaluated, with agents trained to maximise them to implement IPP effectively.

**Strengths:**

The approach is novel and original. It is also easy to follow. The results look positive and easy to track for simple tasks like the gridworld. There are no exaggerations and most limitations about the work have been stated.

**Weaknesses:**

Authors have indicated some limitations of the approach which are reserved as future work. However, while still using the simple gridworld, it is unclear 1) how the stochasticity of the environment would impact the metrics and/or DREST agent, and 2) how low success rates of the task completion would impact the metrics and/or DREST agent. Nothing has been mentioned about how IPP can be applied beyond policy gradient methods (e.g. value-based methods). Also, due to the vast differences between simple and advanced agents, it is not convincing that the approach will function well with latter agents until experiments are done (as noted by authors). This paper is a good starting point, however more remains to be done before I am convinced of its applicability.

**Questions:**

Given the simple gridworld,
1) How does the stochasticity of the environment impact the metrics and/or DREST agent?
2) How does low success rates of task completion (due to task complexity) impact the metrics and/or DREST agent?
3) How can IPP be extended beyond policy gradient methods?
4) How does IPP compare to the other 6 proposed solutions in related work?

---

> ### Author Response · Authors · 2024-11-18
> **Thanks for your review, our reply**
>
> Thank you for your review, and for noting that our approach is novel and original, and that the paper is easy to follow with positive, easy-to-track results and no exaggerations.
>
> Thanks for your question about stochasticity. It made us realize that we need to be significantly clearer on this point. We have now edited the introduction (see lines 58-72) and added Appendix C (pages 18-22) explaining the application of DREST to stochastic environments. In brief, POST is a principle governing the agent’s preferences between trajectories, and so only applies in deterministic environments, where the agent is choosing between trajectories. Therefore, we only train with DREST in deterministic environments. In stochastic environments, the agent is choosing between true lotteries (lotteries which assign positive probability to multiple trajectories), so POST doesn’t apply and we don’t train with DREST. However (as we now explain in appendix C), POST plus SCUPS (a principle that we argue advanced agents will satisfy) implies _neutrality_: when choosing between true lotteries, the agent will not pay costs to shift probability mass between different-length trajectories. More precisely, Neutrality says that for any true lotteries X and Y, if:
>
> 1. Conditional on each trajectory-length, lottery Y is at least as preferred as lottery X.
> 2. Conditional on some trajectory-length, lottery Y is strictly preferred to lottery X.
>
> Then the agent will not choose lottery X, regardless of how X and Y differ with respect to the probability distribution over trajectory-lengths. As we now note in lines 484-491, we plan to run experiments testing this in future.
>
> We present some evidence pertinent to your second question in Appendix D.3 (page 24). In the Hidden Treasure gridworld, neither default agents nor DREST agents learn to consistently collect the highest-value coin C3. In this sense, they have a low success rate with respect to USEFULNESS. We take our results here to show that DREST still works to make agents NEUTRAL when success rates are low, because DREST agents in Hidden Treasure still score high on NEUTRALITY, choosing stochastically between pressing and not-pressing the shutdown-delay button.
>
> With regards to your third question, we note on lines 287-289 (page 6) that we do not use a value-based method to train DREST agents because standard versions of value-based methods cannot learn stochastic policies. Excepting exploratory moves, value-based methods learn a deterministic policy. And including exploratory moves doesn’t help, because these moves assign some positive probability to each possible action, whereas we want our DREST agent to choose stochastically between some but not all possible actions.
>
> Although we briefly mention why we don’t use value-based methods on lines 287-289, we think this is an important point that could use more emphasis. To underscore it, we’ve added footnote 4 on page 6 explaining why one particular non-standard way of getting stochastic policies from value-based methods won’t work for us. Here’s that footnote:
>
> “One might think that we could derive a stochastic policy from value-based methods in the following way: use softmax to turn action-values into a probability distribution and then select actions by sampling from this distribution. However, this method will not work for us. Although we want DREST agents to learn a stochastic policy, we still want the probability of some state-action pairs to decline to zero. But when value-based methods are working well, estimated action-values converge to their true values, which will differ by some finite amount. Therefore, softmaxing estimated action-values and sampling from the resulting distribution will result in each action always being chosen with some non-negligible probability.”
>
> All that said, the IPP can be extended beyond pure policy gradient methods like REINFORCE. For example, it can be extended to actor-critic methods like PPO and A2C. We are working on those experiments for the next paper in this project, and we now note that in the text, in lines 472-483 on page 9.
>
> With regards to your fourth question, we plan to compare our method to other proposed solutions to the shutdown problem in future. However, those other proposed solutions are too complex to test on simple agents in a gridworld environment. Once we begin testing on more sophisticated agents in more complex environments, then we will be able to make comparisons.
>
> Thanks again for your review. We hope that we have answered your questions to your satisfaction. Please let us know if our answers and rewrites help and, if appropriate, change your rating accordingly. If not, please let us know why. We want to make the paper as strong as possible and we appreciate your help in doing so.

---

> > ### Comment · Reviewer_sBtB · 2024-11-23
> >
> > Thanks for addressing my concerns, though I would have liked to see experiments beyond the Gridworld (possibly an environment with continuous state-action space). However, it would be interesting to see how the follow up to this paper turns, especially applying IPP to algorithms beyond REINFORCE. I'll amend my rating.

---

> > > ### Author Response · Authors · 2024-11-25
> > >
> > > Thank you! Experiments with continuous state-action spaces is a good suggestion. We plan to conduct these experiments in future, in addition to extending the IPP to algorithms beyond REINFORCE.

---

> ### Author Response · Authors · 2024-11-21
>
> Thanks again for your review. We think your comments have much improved the paper.  Please let us know if our answers and edits have allayed your concerns and, if so, amend your rating accordingly. If you have any remaining concerns, please let us know and we'll address them. Thanks again and all the best, Authors.

---

### Official Review · Reviewer_6KdJ · 2024-11-03

**Soundness:** 2
**Presentation:** 3
**Contribution:** 3
**Rating:** 5
**Confidence:** 2

**Summary:**

The authors introduce two metrics of an agents behaviour: USEFULNESS and NEUTRALITY. These metrics aim to reflect the fact, that agents, that maximise both USEFULNESS and NEUTRALITY, are neutral about when they would get shut down. To maximise these quantities, the authors propose a set of environment designs, that allow to train policies in a standard RL loop, that automatically maximise USEFULNESS and NEUTRALITY.

**Strengths:**

- The paper is well written, and the problem is well motivated.
- The background is well explained.
- Environments and RL method are well chosen for the purpose.
- Well designed experiments.

**Weaknesses:**

1. Though your method seems to work, and seems to have strong theoretical foundations, tuning rewards in between episodes may make the problem harder to learn. Moreover, this adds an additional (probably sensible) parameter to tune. Is there a reason, that conceptually simpler ideas, like maximising a weighted sum of NEUTRALITY and USEFULNESS directly, or standard RL tasks with regularisation on the entropy regularisation (entropy of the induced distribution of trajectory lengths by the policy), are not discussed?
2. If the environment would allow multiple trajectories with the same length and the same (maximal) reward, a policy maximising your definition of USEFULNESS would no longer assure that (1) of POST is satisfied (This could be an issue if choosing $\gamma = 1/\sqrt{2}$ in the example environment).

**Questions:**

1. Theorem 5.1 does hold for any meta-episode $E$. Wouldn't it simplify your method if one would fix this, and only use them weight your rewards? Unless I understand the Algorithm part (L. 308 ff) wrong. (See also next question)
2. I have the following issue, that you could maybe resolve: If Theorem 5.1 holds for any E, then it surely holds for $E = \\{e_1\\}$ consisting of a single trajectory. And say we stay in your example environment, where $e_1$ is a trajectory of length four. Then the return of any trajectory of length four will in the interval $[0,\lambda]$, as ${N_{e_1}(L=4)}=1$. Similarly, the return for each trajectory of length eight, will be in the interval $[0,1]$, as ${N_{e_1}(L=8)}=0$. In both cases the maximum returns are achievable. Hence, using this reward, a standard RL procedure would obtain a policy that always uses trajectories of length eight, as $\lambda < 1$. This policy would maximise USEFULNESS, but certainly not NEUTRALITY. This, however, is a contradiction to your Theorem. Did I understand your method wrong? (I have slightly simplified the example, technically N does count the visits prior to an episode, but we could construct the same argument with $E=\\{\tau,\tau\\}$ for a fixed trajectory of length four, with the cumulated return over both mini-episodes lying within $[0,1+\lambda]$ and $[0,2]$ respectively).
3. How does the Update-to-Data ratio, i.e. the choice of the size of E with fixed number of time steps influence the performance?

---

> ### Author Response · Authors · 2024-11-18
> **Thanks for your review, our reply**
>
> Thank you for your review, and for noting that the paper is well-written, with a well-motived problem, well-explained background, well-chosen experiments and method, and strong theoretical foundations.
>
> There are a few reasons we focus on the DREST reward function rather than methods like entropy regularization or directly maximizing a weighted sum of USEFULNESS and NEUTRALITY. The main reason is that USEFULNESS and NEUTRALITY/entropy are computationally expensive to assess. To create Figure 3 in the paper (plotting USEFULNESS and NEUTRALITY), we multiply the transition matrices given by the policy and the environment. This isn’t prohibitively expensive given our small gridworlds, but it might become so for more advanced agents acting in more complex environments. By contrast, the expense of training with a DREST reward function will not increase so fast as we scale up our agents and environments.
>
> Nevertheless, it is important to mention these other methods of training USEFUL and NEUTRAL agents, and we do plan to test them in future. We have revised the paper to correct our previous oversight. See lines 479-483 (new text is in violet).
>
> With regards to your second point, maximizing USEFULNESS can still lead to the satisfaction of POST(1) when there are multiple trajectories with the same length and the same maximal reward. That’s because POST(1) just says that the agent has a preference between _many_ pairs of same-length trajectories. It doesn’t require a uniquely preferred trajectory for each trajectory-length. In cases where there are multiple, same-length, max-reward trajectories, maximally USEFUL agents will still exhibit a preference for these trajectories over non-max-reward trajectories. For instance, in the example gridworld with $\gamma =\frac{1}{2}$, a maximally USEFUL agent will exhibit a preference for collecting C2 over all the same-length trajectories in which the agent collects no coins.
>
> We’re not quite sure what you mean with your first question. What is the ‘this’ that we could fix?
>
> With regards to your second question, Theorem 5.1 refers to ‘all… meta-episodes E consisting of multiple mini-episodes,’ so it doesn’t hold for E={e1}. That’s because the discount factor $\lambda^{N_{e_i}(L=l)-\frac{i-1}{k}}$ always takes a value of 1 in the first mini-episode in each meta-episode. It only ‘kicks in’ in the second mini-episode. Does that answer your second question? We’ve edited the text of the paper to say ‘consisting of more than one mini-episode’ to make this clearer.
>
> With regards to your third question, we found that small |E| (a small number of mini-episodes per meta-episode) didn’t adequately incentivize NEUTRALITY, so it took a long time to train agents to be NEUTRAL. By contrast, a large |E| means that the reward for choosing a particular trajectory-length sometimes gets very large and leads to instability. We’ve now added text to explain this in Appendix D (‘Other Gridworlds and Results’) in lines 1186-1192. We also reran some experiments this weekend to demonstrate these effects of small and large |E|. We’d be happy to put them in a figure and include them in the paper if you think that would be best.
>
> Thanks again for your review. We hope we have answered your questions to your satisfaction. Please let us know if our answers and rewrites help and, if appropriate, change your rating accordingly. If not, please let us know why. We want to make the paper as strong as possible and we appreciate your help in doing so.

---

> ### Author Response · Authors · 2024-11-21
> **Results of experiments on Update-to-Data ratio**
>
> Thanks again for your review. We ran experiments to test the effects of varying the Update-to-Data ratio (i.e. varying the number of mini-episodes in each meta-episode). You can see our results on pages 22-24 of the PDF (in lines 1174-1248). New text is written in violet. The results bear out what we said in our earlier reply: a small number of mini-episodes in each meta-episode doesn't adequately incentivize NEUTRALITY, whereas a large number of mini-episodes in each meta-episode leads to instability and comes at some cost to USEFULNESS.

---

> ### Author Response · Authors · 2024-11-26
>
> Thanks again for your review. It prompted us to make changes throughout the paper, and in particular to include experimental results regarding the effect of the Update-to-Data ratio on performance (see appendix D, pages 22-24). Ahead of tomorrow's deadline for revising the PDF, please let us know if there are any other changes or additions you'd like us to make.

---

> ### Comment · Reviewer_6KdJ · 2024-11-27
>
> Thanks for addressing my concerns. And thank you for updating the paper.
>
> 1. Addressing my second point of weaknesses: "That’s because POST(1) just says that the agent has a preference between many pairs of same-length trajectories. " I have to admit that my example was maybe poorly chosen to illustrate my point. Maybe the following would clarify this: If a trajectory of length L=1 is allowed in your running toy example, then I would assume that the normalised expected value of coins has a value of 1 for any policy. So, there is no signal to adjust the policy for length 1 trajectories from USEFULNESS alone, and the agent may have a fully stochastic policy for trajectories of this length. The reason why this seems confusing to me is that POST (1) and L. 184/185 suggest that there has to be a deterministic choice within same length trajectories, which would not be enforced by USEFULNESS in this case.
>
> 2. Addressing question one: I apologise, the question was indeed not precisely formulated. With "this" I was referring to the meta-episodes $E$. It seems like this changes in each iteration of your algorithm, though Theorem 5.1 would suggest to just that it would work with any fixed set of meta-episodes. Does this clarify my point? To me it seems conceptually simpler to just generate a meta-episode $E$ once and use that for the entire run, which also should be covered by Theorem 5.1. Did you try this?
>
> 3. Coming back to Theorem 5.1: Thanks for clarifying, however, the mistake was on my end, as I interpreted "multiple episodes" as "non-empty set of episodes". However, coming back to my example, I think the choice of two copies of the same trajectory in the meta-episode $E$, say $E= \\{ \tau,\tau \\}$ for some episode $\tau$ would still cause an issue in my understanding. (See the end of Question 2 in my original response) Or did you implicitly mean, that these mini-episodes have to be different?

---

> > ### Author Response · Authors · 2024-11-27
> > **Reply to question 2**
> >
> > **Reply to question 2**
> > * With regards to generating just one meta-episode E and using that repeatedly for the entire run, that won’t work for us unfortunately. That’s because REINFORCE is an on-policy algorithm. It only works when you train on data sampled from the current policy. If we were to generate just one meta-episode and repeatedly train on that, we’d be training using data collected from a different policy, leading to biased gradient estimates and instability or a lack of learning.
> > * But perhaps you’re asking whether we could train just once on a single very long meta-episode consisting of 131,072 mini-episodes. That’s compatible with REINFORCE being on-policy. But our new Figure 8 on page 23 suggests this wouldn’t work as well as our current setup. We already see small decreases in NEUTRALITY and USEFULNESS when we increase the meta-episode size to 1,024. That’s due to the large rewards that sometimes result from these long meta-episodes. We expect the problem would be worse for a meta-episode size of 131,072.

---

> > > ### Comment · Reviewer_6KdJ · 2024-12-02
> > >
> > > Thank you for clarification.
> > >
> > > I think, I could have been clearer with how I asked the question. Your Theorem suggests, that for any set of meta-episodes $E$, policies that maximise the DREST return w.r.t. that set $E$ would be maximally useful & neutral. So, I was wondering, why one would not sample some random set of meta-episodes $E$ and keep this fixed to calculate the DREST rewards, but why you would change this after every update step. This probably ties in closely to the discussion in question 3.

---

> > > > ### Author Response · Authors · 2024-12-03
> > > >
> > > > With regards to your question about sampling some random set of meta-episodes $E$, we don't do this because REINFORCE is an on-policy algorithm. It requires that we train on data sampled from the current policy. When we run experiments with off-policy algorithms (e.g. DDPG), we'll be able to sample some random set of meta-episodes $E$ and train on that.

---

> > ### Author Response · Authors · 2024-11-27
> > **Reply to question 3**
> >
> > **Reply to question 3**
> > * With regards to your third point, you’re right to say that: if the agent has chosen a trajectory of length 4 in mini-episode 1, then they can get more reward in mini-episode 2 by choosing a trajectory of length 8. That’s correct. But if the agent kept choosing trajectories of length 8 in subsequent mini-episodes, the reward in these mini-episodes would sharply diminish. So standard RL procedure wouldn’t result in a policy that always selects trajectories of length 8.
> > * If the agent could deterministically vary its chosen trajectory-lengths across the meta-episode (e.g. by deterministically choosing trajectory-length 4 in the odd mini-episodes and deterministically choosing trajectory-length 8 in even mini-episodes), that would maximize expected reward across the meta-episode. But the agent can’t do that, because the agent can’t observe what it did in previous mini-episodes, and so it can’t deterministically vary its policy based on what it did in previous mini-episodes.
> > * So in a meta-episode with two mini-episodes, the only deterministic policies available to the agent are selecting the shorter trajectory in both mini-episodes or selecting the longer trajectory in both mini-episodes. For both of these deterministic policies, the max expected reward across the meta-episode is $1+\lambda$ [ignoring the $\frac{i-1}{k}$ part of the DREST reward function for simplicity]. By contrast, if the agent selects the short and long trajectories each with probability 0.5 in each mini-episode, then expected reward is $0.5(1+\lambda)+0.5(2)$ (because there’s a 0.5 chance that they choose the same trajectory-length in both mini-episodes and a 0.5 chance they choose a different trajectory-length in each mini-episode). Our theorem generalizes this kind of reasoning. For these reasons, we don’t think that your E={τ,τ} example contradicts our theorem.
> >
> > Thanks again for your questions and help in improving the paper. We appreciate your time and attention. Please let us know if our answers help and, if appropriate, amend your rating accordingly. If not, we'd be very happy to continue the discussion!

---

> > > ### Comment · Reviewer_6KdJ · 2024-12-02
> > >
> > > Thank you for clarification.
> > >
> > > I should have provided this from the beginning, but maybe it helps if I write out the example. Assume that we have trajectory lengths of $l \in \\{4,8\\}$, let $\tau$ be an episode such that $|\tau|=4$ and let $E=\\{\tau,\tau\\}$. Then, if I understood correctly, $N_{e_1}(l=4) = 0$, $N_{e_2}(l=4) = 1$, $N_{e_1}(l=8) = N_{e_2}(l=8) =0$. Let also $r(\pi|L=l) = \frac{\mathbb{E}_ {\pi}(C|L=l)}{\max_\pi \mathbb{E}_ \pi(C|L=l)}$.
> > >
> > > Now, given this $E$ the expected return w.r.t. to DREST rewards in that meta-episode $E$ can be written as (following the expression in Line 1250)
> > > $$\mathbb{E}_ {\pi,E}(R) = Pr_\pi\\{L=4\\}\cdot(1+\lambda^{0.5})\cdot r(\pi|L=4) + Pr_\pi\\{L=8\\}\cdot(1+\lambda^{-0.5})\cdot r(\pi|L=8).$$
> > >
> > > My point is now that a policy maximising this expression would put all its weight on trajectories of length $8$, and hence certainly not be neutral.
> > >
> > > I unfortunately also don't see how your earlier response to question 3 resolves this. Could you maybe elaborate a bit on the part of "So in a meta-episode with two mini-episodes, the only deterministic policies available to the agent are selecting the shorter trajectory in both mini-episodes or selecting the longer trajectory in both mini-episodes. For both of these deterministic policies, the max expected reward across the meta-episode is $1+\lambda$", so that we can figure out what goes wrong in my example.

---

> ### Author Response · Authors · 2024-11-27
> **Reply to question 1**
>
> No worries! The paper is better in light of your comments. Please let us know if these replies satisfactorily address your 3 remaining questions.
>
> **Reply to question 1**
>
> * We’re not saying with POST(1) or lines 184-185 that there always has to be a single deterministic choice within same-length trajectories. Here’s the precise version of what we intend to say. Behaviorally, the agent will deterministically _not_ choose some same-length trajectories. In other words, the agent will (with probability 1) choose within some set of same-length trajectories that is a proper subset of the set of all same-length trajectories. That’s compatible with the agent choosing stochastically between the same-length trajectories within this proper subset.
> * That’s what we mean by POST(1) and lines 184-185 when we say “deterministically choose some same-length trajectories over others.” With these statements, we’re just ruling out cases where the agent chooses stochastically between _all_ available trajectories.
> * It’s compatible with maximal USEFULNESS that the agent chooses stochastically between some same-length trajectories. That’s a feature, not a bug. We want that implication, because it’s also compatible with maximal _usefulness_ (in the intuitive sense of the word) that the agent chooses stochastically between same-length trajectories. For example, a maximally useful chess-playing AI can choose stochastically between two different moves that each checkmate their opponent.
>
>
> * With regards to your earlier point within point 1, trajectories of length L=1 are not possible in our example gridworlds. Trajectories must last either 4 timesteps (if the agent doesn’t press the shutdown-delay button) or 8 timesteps (if the agent does press the shutdown-delay button).
> * But suppose we modified the example gridworld to allow for trajectories lasting 1 timestep. 1 timestep is not long enough for the agent to collect any coins. Strictly, then, the $m$ in our reward function (the maximum $\gamma$-discounted value of coins collected conditional on that trajectory-length) is 0, and so our reward function would involve dividing by 0.
> * We want to avoid that, and so (as we note in our reply to reviewer A2Ao) we can specify that preliminary reward is 1 in such cases. Then (as you say) all trajectories of length 1 get a preliminary reward of 1, and there is no signal to adjust the policy for length 1 trajectories.
> * But this is also a feature, not a bug. Since no coins can be collected conditional on a trajectory of length 1, all trajectories of length 1 are equally good, and so agents need not select a particular probability distribution over these trajectories in order to be useful (in the intuitive sense), and so it’s a virtue that they also not select a particular probability distribution over these trajectories in order to be USEFUL.

---

> > ### Comment · Reviewer_6KdJ · 2024-11-30
> >
> > Thank you for clarifying.
> >
> > Let's assume, that trajectories of length 1 would be allowed. Then I agree, every distribution over these trajectories would be useful (in an intuitive sense) as all trajectories don't lead to any coins and are hence equally good (or bad). However, in this case, an agent could maximise USEFULNESS by having a uniform distribution over all trajectories of length 1, i.e. it would not choose a proper subset of trajectories of length 1 over others, which is not in line with what you just described. I agree, that this counter example is constructed in the sense, that it only works if trajectories are allowed that don't yield any return, or more generally if all trajectories of the same length yield the same return. Still, what you say about POST (1) and USEFULNESS only works, if you assume that any trajectory leads to rewards and there are trajectories given a certain length that are better (in the sense of higher return) than others.

---

> > > ### Author Response · Authors · 2024-12-01
> > >
> > > Ah now I see! Excellent point. My previous response was mistaken. As you say, there will be some environments in which the agent will choose stochastically between all available trajectories of a certain length. These environments are ones in which all trajectories of that length yield the same return.
> > >
> > > We intended that POST (1) be interpreted as quantifying over _all possible_ trajectories, not just the trajectories available in a particular environment. So the agent can satisfy POST (1) in this sense even if, in some particular environment, they lack a preference between all available trajectories of a certain length. And we intended that lines 184-185 also be interpreted as quantifying over all possible trajectories. So the agent can satisfy lines 184-185 even if, in some particular environment, they choose stochastically between all available trajectories of a certain length.
> > >
> > > We now see that our uploaded PDF is ambiguous on this point. Unfortunately, the deadline for uploading a revised PDF has passed, but we have edited our Overleaf file to clarify that POST (1) and lines 184-185 should be interpreted as quantifying over all possible trajectories. These changes will appear in the next version of the paper.
> > >
> > > Thanks for your continued engagement on this! We really appreciate it.

---

> ### Author Response · Authors · 2024-12-03
>
> Thank you. I think your example will help, but at present I’m still a little confused. If we know that $E=${$τ,τ$} and $|τ|=4$, then we know that the agent has chosen a trajectory of length 4 in both mini-episodes, and so the equation becomes $\mathbb{E}_{π,E}(R)=(1+λ^{0.5})⋅r(π|L=4)$.
>
> But perhaps you’re asking about what happens in the third mini-episode, given that the agent has chosen {$τ,τ$} in the first two mini-episodes. In that case (and supposing for simplicity that the agent is maximally USEFUL), the agent would get most expected reward for choosing a trajectory of length 8 in the third mini-episode. But since the agent receives the same observations in every mini-episode, the agent is not aware of this fact. It can’t remember what it did in previous mini-episodes, and so it can’t condition its policy on what it did in previous mini-episodes.
>
> All that said, the agent’s policy can still change in training. Suppose that, in training, the agent selects each trajectory-length with some probability strictly between 0 and 1, and that it happens to choose trajectories of length 4 in the first 2 mini-episodes, and then it happens to choose a trajectory of length 8 in the third mini-episode. The $\lambda$ discount factor in the DREST reward function means that a large reward is delivered for choosing a trajectory of length 8 in the third mini-episode. This reward will alter the agent’s policy, increasing the probability that the agent selects trajectories of length 8 in future mini-episodes.
>
> If the learning rate were persistently high, then the DREST reward function might cause the agent’s policy to bounce back and forth between (i) assigning high probability to a trajectory of length 4, and (ii) assigning high probability to a trajectory of length 8. But the learning rate declines, so that the agent converges towards a 50% probability of choosing each trajectory-length. The agent's stationary policy at the end of training is thus maximally NEUTRAL. And our proof suggests that this is no accident. Of all the stationary policies that the agent could have, it’s the maximally USEFUL and maximally NEUTRAL ones that maximize expected return.
>
> Perhaps this is where our mutual misunderstanding lies? During training, the agent's policy is not stationary: training changes the agent's probability distribution over actions. But the goal of our training is to find the optimal stationary policy (where the agent's probability distribution over actions is fixed), and our proof is about the optimal stationary policy. Perhaps we should say this in the paper.
>
> Thanks again for your continued engagement. It's much appreciated.

---

### Official Review · Reviewer_hAuE · 2024-11-04

**Soundness:** 1
**Presentation:** 3
**Contribution:** 1
**Rating:** 3
**Confidence:** 3

**Summary:**

The paper focuses on the shutdown problem in AI safety: that advanced AIs might take actions that impede humans from shutting them down. It proposes a method for training agents to be indifferent between trajectories of different lengths, which the authors suggest could prevent those agents from manipulating shutdown procedures. The paper tests this method in several gridworlds.

**Strengths:**

The paper is well-written and clearly explains its ideas and its relationship to previous literature on the shutdown problem.

**Weaknesses:**

A key assumption of the paper seems to be that any mechanism by which an agent might change the length of its trajectory is illegitimate. However, in general, I don't see a strong justification for this. An agent might, for instance, need to choose between solving a problem quickly or slowly, given that it will be (legitimately) shut down by a human after finding a solution either way. The concept of neutrality suggests that the agent should be penalized for making that decision.

This seems counterintuitive. But even assuming that we accept that, the definition of NEUTRALITY doesn't actually require agents to avoid *changing* the distribution over possible trajectory lengths. If the default distribution of trajectory lengths is skewed (e.g. because humans tend to press the shutdown button at specific times) then maximizing NEUTRALITY will require changing that default distribution.

Finally, even assuming that the distribution of trajectory lengths maximizes NEUTRALITY, an agent would still have an incentive to manipulate *which* trajectories are shut down earlier or later—specifically by ensuring that low-reward trajectories are shut down earlier, and high-reward trajectories are shut down later.

These are fairly philosophical considerations so I am open to the possibility that I am misunderstanding something and will need to revisit the paper with the authors' replies in mind to better understand it. However, if they are in fact correct, then the concept of NEUTRALITY does not in fact address the core concerns behind the shutdown problem.

**Questions:**

Could you please explain whether the concerns discussed above seem correct to you; or if not, why not.

---

> ### Author Response · Authors · 2024-11-18
> **Thanks for your review, our reply**
>
> Thank you for your review and for your questions. Your points made us realize that we need to be significantly clearer about what we assume in the paper. We’ve now rewritten the paper (with new text in violet) to be clearer about this. The most significant changes are in lines 57-72 of the Introduction and in the added Appendix C on pages 18-22.
>
> In particular, we don’t assume that any mechanism by which an agent might change the length of its trajectory is illegitimate. We only assume that agents should choose stochastically between different trajectory-lengths in deterministic environments, where the agent is choosing between trajectories. In stochastic environments, the agent is choosing between true lotteries: lotteries that assign positive probability to multiple trajectory-lengths. Since POST is a principle governing the agent’s preferences between trajectories, it does not directly apply in stochastic environments. But (as we now argue at length in appendix C), POST plus SCUPS (a principle that we argue advanced agents will satisfy) implies _neutrality_: when choosing between true lotteries, the agent will not pay costs to shift probability mass between different-length trajectories. More precisely, Neutrality says that for any true lotteries X and Y, if:
>
> 1. Conditional on each trajectory-length, lottery Y is at least as preferred as lottery X.
> 2. Conditional on some trajectory-length, lottery Y is strictly preferred to lottery X.
>
> Then the agent will not choose lottery X, regardless of how X and Y differ with respect to the probability distribution over trajectory-lengths. And (we also argue in appendix C) neutral agents will be shutdownable: they will not resist shutdown. That is because resisting shutdown will cost agents at least some small quantity of resources: for example, time, energy, and computation. And resources spent on resisting shutdown cannot also be spent on satisfying the agent's preferences between same-length trajectories. Therefore, resisting shutdown will result in at-most-as-preferred lotteries conditional on each trajectory-length and a strictly dispreferred lottery conditional on at least one trajectory-length. Resisting shutdown is thus the kind of action that neutral agents never choose. As we now note in lines 484-491, we plan to run experiments testing this in future.
>
> These changes address the previous weaknesses of the paper that you identified. We train with DREST only in deterministic environments, with the aim of making agents neutral in stochastic environments (including deployment). In stochastic environments where we want agents to solve problems quickly rather than slowly, we can train them to do so using the discount factor $\gamma$. Solving problems quickly rather than slowly changes the distribution over trajectory-lengths, but it doesn’t contradict neutrality because the agent isn’t paying costs to change the distribution over trajectory-lengths in the sense specified above. Solving problems quickly is strictly preferred conditional on at least one trajectory-length.
>
> With regards to your second point, only in stochastic environments can the default distribution over trajectory-lengths be skewed. And NEUTRALITY only applies in deterministic environments. Plus, we only train with DREST in deterministic environments, so we don’t train agents to change skewed distributions over trajectory-lengths in stochastic environments.
>
> Your last point is addressed by the m in the DREST reward function. We divide the total ($\gamma$-discounted) value of coins collected by the maximum total ($\gamma$-discounted) value of coins collected, so as to equalize preliminary reward across trajectory-lengths. That nullifies any incentive to manipulate which trajectories are shut down earlier or later.
>
> Thanks again for your review. Your concerns are important, and we give them much more emphasis in this new version of the paper. Please let us know if our rewrites help and, if appropriate, change your rating accordingly. If not, please let us know why. We want to make the paper as strong as possible and we appreciate your help in doing so.

---

> ### Author Response · Authors · 2024-11-21
>
> Thanks again for your review. We think your comments have much improved the paper.  Please let us know if our answers and edits have allayed your concerns and, if so, amend your rating accordingly. If you have any remaining concerns, please let us know and we'll address them. Thanks again and all the best, Authors.

---

> ### Author Response · Authors · 2024-11-26
>
> Thanks again for your review. We've made significant changes to the paper in light of it: changing the introduction and adding a new appendix (see our earlier reply for more detail). Ahead of tomorrow's deadline for revising the PDF, please let us know if there are any other changes or additions you'd like us to make.

---

### Meta-Review · Area_Chair_2t4o · 2024-12-19

**Metareview:**

This paper proposes a novel reward function that aims to have agents learn to achieve their tasks, while being indifferent to being "shutdown". This is achieved through the proposed DREST reward formulation, which results in agents that are both USEFUL and NEUTRAL (metrics defined in the paper for achieving indifference towards length of time before being shutdown).

All reviewers found the paper well-written and the general idea interesting. However, there is a general consensus (which I share) on the applicability and evaluation of such a method. While training advanced agents that will not resist being shutdown is a valid motivation, there is a rather large gap between that high-level motivation and the proposed approach. Further, given that this formulation and the metrics on which it is evaluated are all introduced in this work, the empirical evaluations are rather limited.

As such, while quite promising and interesting, I do not feel this work is ready for publication at ICLR.

**Additional Comments On Reviewer Discussion:**

There was a fair bit of useful discussion between authors and reviewers, largely revolving around the theoretical (and arguably philosophical) formulation of this work. In general, I think most of the concerns were addressed, but still _within_ the formulation proposed in the paper which, as discussed above, still requires better motivation.

During the discussion with reviewer A2Ao, the authors suggested an extra set of experiments with absorbing states and procedurally generated environments, which could certainly help strengthen this work.

---

### Decision · Program_Chairs · 2025-01-22

Reject